# MAPCap allows high-resolution detection and differential expression analysis of transcription start sites

Vivek Bhardwaj [1,2,3], Giuseppe Semplicio[1,3], Niyazi Umut Erdogdu [1,2], Thomas Manke [1] & Asifa Akhtar [1]

The position, shape and number of transcription start sites (TSS) are critical determinants of gene regulation. Most methods developed to detect TSSs and study promoter usage are, however, of limited use in studies that demand quantification of expression changes between two or more groups. In this study, we combine high-resolution detection of transcription start sites and differential expression analysis using a simplified TSS quantification protocol, MAPCap (Multiplexed Affinity Purification of Capped RNA) along with the software icetea. Applying MAPCap on developing *Drosophila melanogaster* embryos and larvae, we detected stage and sex-specific promoter and enhancer activity and quantify the effect of mutants of maleless (MLE) helicase at X-chromosomal promoters. We observe that MLE mutation leads to a median 1.9 fold drop in expression of X-chromosome promoters and affects the expression of several TSSs with a sexually dimorphic expression on autosomes. Our results provide quantitative insights into promoter activity during dosage compensation.

[1] Max Planck Institute for Immunobiology and Epigenetics, 79108 Freiburg, Germany. [2] Faculty of Biology, University of Freiburg, 79104 Freiburg, Germany. [3] These authors contributed equally: Vivek Bhardwaj, Giuseppe Semplicio. Correspondence and requests for materials should be addressed to A.A. (email: akhtar@ie-freiburg.mpg.de)

Most genes in eukaryotes produce multiple transcript isoforms, contributing to tissue-specific regulation of gene expression. Isoform diversity can be achieved by the usage of alternative exons, untranscribed regions (UTRs) and transcript start and end sites. Recent analysis of human genome suggests that transcript start and end site selection is a major driver of alternative isoform usage across tissues[1].

Selection of alternative transcription start sites (TSSs) also reflects a change in promoter usage of a gene. Promoter-profiling methods, such as CAGE[2], RAMPAGE[3], NanoCAGE[4] and GRO-cap[5], allow a genome-wide detection of promoter usage by identification of TSSs. A recent comparative analysis of six such methods identifies the latest, PCR-free variant of CAGE (nAnT-iCAGE) as the overall best method in terms of accuracy of detected TSSs, while RAMPAGE[3] comes a close second[6]. The amount of time and number of steps required per library was found to be highest for CAGE. RAMPAGE reduces the processing time (to 2 days) and the required input material (down to 5 µg), therefore providing a suitable alternative. The challenge of further reducing the input material have been tackled in protocols such as nanoCAGE[4] and SLIC-CAGE[7], which require only nanograms of RNA, while the latest C1 CAGE protocol[8] allows detection of TSSs in single cells. These protocols have focussed on increasing detection sensitivity of lowly abundant capped RNAs such as eRNAs[8] and on high-resolution mapping of TSS[7], rather than accurate quantification of their expression. Quantification of transcript expression from CAGE is hindered by the PCR amplification bias. The nAnT-iCAGE protocol removes the PCR amplification step to correct this bias, with a lower limit down to 5 µg of starting material[9]. Protocols such as HeliScopeCAGE[10] and RAMPAGE[3] aim to tackle this by either sequencing unamplified RNA on a single-molecule sequencer (HeliScopeCAGE) or by incorporating pseudo-random barcodes in the reads, followed by post-mapping de-duplication (RAMPAGE). Detection as well as quantification could further be improved by the use of biological replicates and spike-in RNA controls for normalization, a strategy that has proven highly beneficial for RNA-seq studies[11]; however, promoter-profiling studies have so far not employed these strategies. Other potential reasons for the limited scope of promoter-profiling protocols could be the relative difficulty of performing the protocols[6], as well as low correlation of gene-level expression estimates with RNA-seq[12].

In *Drosophila*, promoter profiling has been used to study promoter architecture[13], promoter usage during development[3], relationships between promoter shape and transcription noise[14], detection of enhancer RNAs and divergent transcripts[15,16], and to study conserved non-coding transcription across species[17]. Dosage compensation is another biological phenomenon that could benefit from accurate quantification of promoter activity. In *Drosophila melanogaster*, the imbalance of sex chromosomes between male and female flies is compensated via the transcriptional upregulation of the single male X chromosome[18,19]. This process requires the male-specific lethal (MSL) complex, which is responsible for deposition of H4K16 acetylation on promoters and gene bodies of X-chromosomal genes[20,21]. The sex-specificity of this process relies on the male-specific expression of the MSL complex protein MSL2[22,23], along with a highly male-biased expression of the roX non-coding RNAs on the X chromosome[24]. Another crucial protein in the process is MLE (maleless), which incorporates *roX* RNAs in the MSL complex, facilitating its spread on the X chromosome[25]. Finally, the enzyme MOF (males absent on first) deposits the H4K16ac mark on X, which is associated with hyperactivation of gene expression[26,27]. Although this global upregulation of gene expression has been quantified previously using RNA-seq[28,29] and GRO-seq[30,31], the variability of promoter usage and expression of different transcripts (such as

ncRNAs, eRNAs, etc.) has not been investigated. A high-resolution, quantitative expression profiling approach could therefore provide more insights into the process.

In this study, we have developed a method termed MAPCap (Multiplexed Affinity Purification of Capped RNA) that allows multiplexed processing of samples in about 16 h and produces long, paired-end reads, enabling high-resolution detection of transcription start sites. Synthetically designed random barcodes are used to remove PCR duplicates, and external spike-in controls allow accurate quantification of changes in TSS expression. Furthermore, we present a method for detection of transcription start sites that utilizes variation between biological replicates to improve TSS accuracy. We implement this method along with various processing and analysis options into an easy to use R/Bioconductor package icetea (Integrating Cap Enrichment with Transcript Expression Analysis; https://bioconductor.org/packages/icetea). Applying MAPCap to developing *D. melanogaster* embryos and larvae, we detect stage-specific activities of enhancers as well as sex-specific TSS expression. We apply MAPCap to quantify expression changes in mutants of the MLE RNA helicase, a protein essential for balancing the X-chromosomal gene dosage between males and females[32] and remarkably find a median 1.9-fold downregulation of TSSs on X chromosome upon loss of MLE. Further comparison of male and female wild-type and mutant flies indicate an unexpected potential function of MLE in regulating the expression of sexually dimorphic genes. Moreover, by analysing the variability of expression defects, we find that genes which are least sensitive to the mutation of MLE are located significantly further from the high-affinity sites (HAS) as well as from the boundaries of topological domains (TADs) compared to MLE-sensitive, dosage-compensated genes located on the X chromosome. We propose that the MAPCap protocol, combined with **icetea** analysis would be a highly useful approach for de novo detection as well as differential expression analysis of TSSs and transcript isoforms in all species.

## Results

**MAPCap enriches Capped RNA from multiplexed samples**. We developed MAPCap by combining bead-based affinity purification of mRNA Cap with a recently described RNA library preparation strategy (Fig. 1a, see "Methods"). MAPCap utilizes the "s-oligo", originally developed for genome-wide RNA–protein interaction (FLASH) assays[33]. s-oligo suppresses the frequently found "adapter-dimers" during the ligation-mediated cloning of RNA, a step that is also important for promoter-profiling protocols. The s-oligo is an RNA/DNA chimera, with a 5'-dangling single-stranded RNA made up of 6 ribonucleotides of sample index (XXXXX), sandwiched between 7 random ribonucleotides that serve as Unique Molecular Identifier (UMI; NNXXXXXXNNNNN). The 5'-end of the s-oligo RNA ligates to the 3'-end of target RNA while the 3'-end of the s-oligo forms a DNA duplex, which also serves as a primer for the reverse transcriptase (RT), making the ligated product ready for RT after dephosphorylation. Early multiplexing of samples makes it easier to handle low input samples for the later stages and also reduces technical variability between samples. Abundant RNA species such as small nuclear RNAs (snRNAs) and small nucleolar RNAs (snoRNAs) are selectively degraded by targeted antisense oligos and RNase H, increasing the recovery of other capped RNA species (Supplementary Tables 1–3). PCR amplification (14–18 cycles, depending on input RNA) creates a uniform library with ideal insert sizes of around 150 nt (Supplementary Fig. 1a). Since our RNA fragmentation is based on sonication, we can easily control the insert size to allow sequencing of longer inserts if

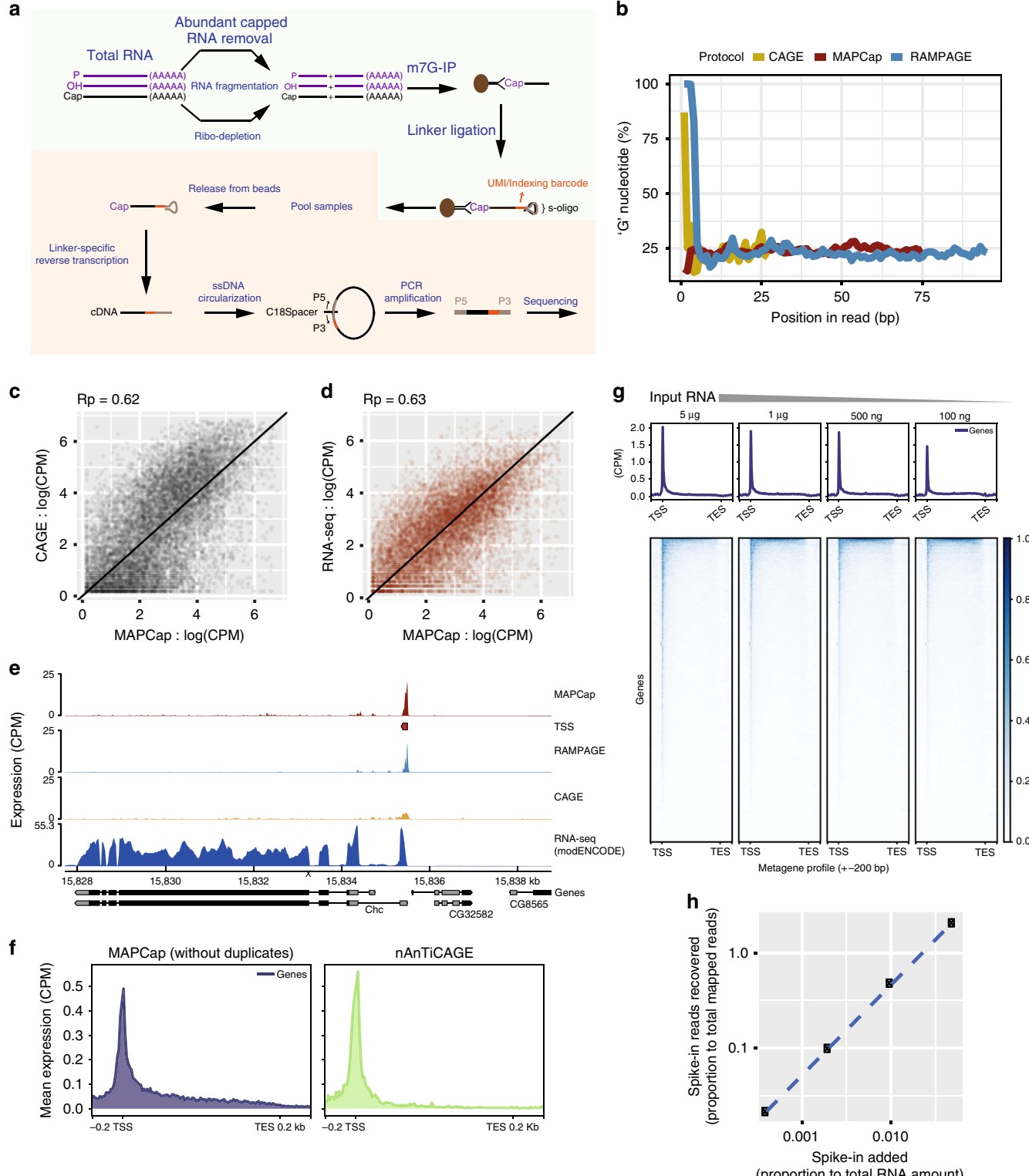

desired, increasing the mapping efficiency. The whole protocol, from RNA isolation to libraries ready for sequencing, takes around 16 h and aims to minimize the critical or difficult steps, making it easy to perform.

To evaluate MAPCap data quality, we performed MAPCap on stage 15 *Drosophila* embryos in four replicates (see "Methods"). We obtained 4.7–8.5 million reads per sample after de-multiplexing (Supplementary Fig. 1b). After mapping and removal of PCR duplicates, the 5'-UTR of genes showed a high correlation of

MAPCap signal between replicates (92–94%, Pearson's $R$, Supplementary Fig. 1c). Similarly, high correlation was observed for the detected TSSs (90–94%, Pearson's $R$, Supplementary Fig. 1d). Similar correlation was observed between replicates of the CAGE data obtained from modENCODE in S2 cells (see "Methods"); however, the correlation reduces after removal of duplicates (Supplementary Fig. 1e, f). For further comparison, we processed the CAGE data downloaded from modENCODE[34] and the RAMPAGE data from embryos corresponding to the same

**Fig. 1** MAPCap (Multiplexed Affinity Purification of Capped RNA) efficiently captures transcript 5'-end signal. **a** Overview of the MAPCap protocol. After fragmentation and ribo-depletion, the Capped RNA is immunoprecipitated using an antibody, and the s-oligos are attached, afterwards the samples are pooled for PCR and library preparation steps. **b** Nucleotide content in read positions. CAGE and RAMPAGE show a high artificial G-bias due to their cloning steps, while MAPCap shows low bias for any specific nucleotide. **c, d** Correlation of signal (log2(counts per million + 1)) between MAPCap and CAGE and between MAPCap and RNA-seq on 5'-untranscribed region of genes. **e** Genome snapshot of the de-duplicated counts from MAPCap, RAMPAGE and CAGE on transcription start site. For MAPCap and RAMPAGE, the de-duplication was performed using 5'-position of the reads and the UMIs, while for CAGE it was performed using only 5'-position. RNA-seq track is shown for comparison. **f** Metagene profile comparing signal enrichment between nAnTiCAGE and MAPCap on all genes in mouse embryonic stem cells (CPM = counts per million). **g** Metagene profile of signal from the MAPCap experiment performed in S2 cells, using different quantity of total RNA (5 μg, 1 μg, 500 ng, 100 ng) as starting material. **h** Added relative concentration of capped spike-ins ((amount of spike-in RNA/amount of total RNA) × 100) vs recovered relative counts ((reads mapped to spike-ins/total mapped reads) × 100) for the embryos. The samples were added with proportionally increasing relative concentration of ERCCs

developmental stage[3]. Cap-trapping and template-switching protocols rely on the enhanced terminal transferase (TdT) activity of the RT during cDNA preparation[35–38]. Therefore, reads obtained from both CAGE and RAMPAGE protocols show a high "G" nucleotide bias, which demands post-mapping correction and can affect the accuracy of TSS detection[39]. MAPCap, on the other hand, shows no such bias. This can be attributed to a highly reduced TdT activity of our RT under protocol conditions (Fig. 1b).

Several differences have been observed in the past between promoter-profiling and the RNA-seq protocol in detection of transcript types as well as gene expression estimates[12,40]. We correlated the depth-normalized counts on 5'-UTRs of known genes between MAPCap, CAGE, RAMPAGE, and the ribo-depleted RNA-seq data from matched developmental stages obtained from the modENCODE project (see "Methods"). We found that, although MAPCap signal shows good correlation with other protocols, it showed a better correlation with RNA-seq (Pearson's R; MAPCap = 0.63, CAGE = 0.48, Fig. 1c, d, Supplementary Fig. 1g–j). This indicated that MAPCap experiments could provide better gene expression estimates than other protocols. PCR amplification has been shown to create a detection and quantification bias in both DNA and RNA-seq, as well as CAGE protocols[10,41]. This bias can be controlled by appropriate removal of PCR duplicates. Random barcodes present in the s-oligos serve as UMIs, allowing us to remove PCR duplicates while preserving the transcript expression signal. Similar de-duplication can be performed for data obtained from the RAMPAGE protocol, where the frequent mismatch between random RT-PCR primers and genome sequence could be treated as "pseudo-random" barcodes[3]. A comparison of PCR duplicate removed signal from the three protocols show that both MAPCap and RAMPAGE preserve the signal on the TSS, while de-duplication in the absence of random barcodes lead to near-complete loss of signal from the CAGE protocol (Fig. 1e, Supplementary Fig. 2a). Taken together, our results suggest that MAPCap produces high-quality quantitative signal at the 5'-end of transcripts.

Although the latest version of CAGE protocol, nAnTiCAGE, does not use PCR amplification[9], the CAGE data used in the modENCODE project does. We therefore performed MAPCap on RNA extracted from mouse embryonic stem cells (ESCs) with a dual-hybrid genotype (*Mus musculus × Mus castenious*) and compared it to the nAnTiCAGE data available online[7] (see "Methods"). MAPCap showed an enrichment of signal at the 5'-UTRs of genes comparable to nAnTiCAGE, along with a 55% correlation of signal (Fig. 1f, Supplementary Fig. 2b, c). Furthermore, we performed allele-specific sorting of signal from our MAPCap data and found that 16% of MAPCap signal could be assigned uniquely to the maternal or paternal allele. This allowed us to detect 173 allele-specific TSSs (94 maternal, 79 paternal, Supplementary Fig. 2e). This initial analysis indicated

that MAPCap also provides suitable coverage to detect allelic bias in TSS usage.

**Sensitivity and accuracy of TSS detection in MAPCap.** In order to test the sensitivity of MAPCap protocol to amounts of starting material, we next performed MAPCap on RNA isolated from *Drosophila* S2 cells, ranging from 5 μg to 100 ng (see "Methods"). Comparison of duplicate-free signal enrichment shows that MAPCap can easily recover signal at TSSs for up to 100 ng of starting material (Fig. 1g). Further, we could detect TSSs using the data at high accuracy for up to 500 ng of starting material (Supplementary Fig. 2e), suggesting that MAPCap can reliably obtain TSSs from up to 500 ng of starting material in the absence of replicates (see next section). We then tested the ability of MAPCap to detect various transcript concentrations. To this end, we prepared a spike-in mix containing 10 in vitro capped ERCC spikes (see "Methods") in a 2-fold relative concentration ranging from 15.6 pM to 8 nM. We then mixed each replicate of the embryo sample (each containing 5 μg total RNA) with different concentrations of this spike-in mix (from 0.0004% to 0.05% of isolated RNA), at the beginning of the protocol. The results showed that the relative concentration of spike-ins between samples can be faithfully recovered after sequencing (Fig. 1h). Relative ratios between individual spike-ins within each mix could also be accurately recovered (Supplementary Fig. 2f).

In order to assess the accuracy of TSS detection of MAPCap compared to other protocols, we detected TSSs from MAPCap data using the paraclu method[42] and compared them to the TSSs detected from CAGE and RAMPAGE with identical read-depth and processing (see "Methods"). We evaluated the TSS detection sensitivity, precision and F1-score between the methods by comparing the TSSs obtained from each method to all annotated TSSs in the *Drosophila* ensembl annotation (release 76), along with the RNA-seq data obtained from modENCODE and the DNAse-seq data obtained from comparable developmental stage[43] (see "Methods"). Since current dm6 ensembl and flybase TSS annotations utilize the data from CAGE and RAMPAGE, we expected these protocols to perform well in these metrics. However, MAPCap showed sensitivity and specificity similar to the other protocols (Supplementary Fig. 3a). Further, about one fifth of the MAPCap TSSs classified as "false positives" showed evidence of promoter activity, such as active histone marks or DNAse hypersensitivity (Supplementary Fig. 3b, see "Methods"). Overall our analysis suggests that MAPCap provides high sensitivity while achieving accuracy comparable to existing standards in detection of TSSs.

**Detection of TSSs using biological replicates improves accuracy.** The ease of use and multiplexing ability of the MAPCap protocol allows performing biological replicates without adding

additional time and effort. We therefore sought to develop analysis methods that could benefit from biological replicates. Commonly used methods for TSS detection are either based on the density of CAGE tags (paraclu[42]) or distance between individual tags (distclu[38]). These methods rely on within-sample clustering of tags and do not incorporate biological replicates to improve the performance of TSS detection. Hence, we developed a window-based TSS detection method similar to the window based peak calling algorithms applied in chromatin immuno-precipitation (ChIP-Seq) analysis[44,45]. Read counts are modelled using negative binomial distribution and the TSSs are detected as windows of enrichment in the genome with respect to a local background (Supplementary Fig. 3c, see "Methods"). Consecutively enriched windows are then merged to detect both sharp and broad TSS shapes. We compared the accuracy of peaks obtained from our "local enrichment" method with those from the popular paraclu method[42] using (1) each replicate alone; (2) reads pooled from all replicates, and (3) intersection of TSS from each replicate. TSSs detected by the local enrichment method using replicates show higher sensitivity as well as specificity (area under precision-recall curve (AUPRC) = 0.838) compared to all these alternatives (AUPRC: 0.57–0.64) (Fig. 2a, b, Supplementary Fig. 3d). In order to test whether this method is generally applicable to other protocols, we applied it on equally subsampled 1 million reads from the CAGE data of the S2 cells from modENCODE project. We found that, although the optimal precision and sensitivity stays similar to the traditional (paraclu) approach, our method detects a larger number of true positive peaks (Supplementary Fig. 3e, f). These results suggest that the replicate-based analysis increases sensitivity and robustness of TSS detection in MAPCap as well as other promoter-profiling methods.

We then evaluated whether the shape of detected TSSs by our method represents a biologically informative signal. Earlier analysis of CAGE data indicated that the genes with sharp, focussed TSSs are mostly tissue specific and developmentally regulated, while the genes with broad TSSs have housekeeping functions[46]. Gene Ontology (GO) term enrichment analysis of the sharp and broad TSSs obtained using the "local enrichment" method confirmed previous results (see "Methods"). Genes with sharp TSSs were enriched for processes like morphogenesis and development while genes with broad TSSs denoted mechanisms such as protein localization, metabolism and membrane organization (Fig. 2c), suggesting that TSS shapes detected from MAPCap reflect known biological signals. Motif enrichment analysis of sharp and broad TSSs further confirm these results, with sharp TSSs frequently containing the initiator (Inr) element, while broad TSSs were enriched for core promoter motifs Motif-1 and 2 (Fig. 2d, Supplementary Fig. 3g, see "Methods"). Known developmental genes such as hunchback (hb) and housekeeping genes such as proteosomal subunit Prosbeta5 also showed the expected TSS shapes (Fig. 2e, f, Supplementary Fig. 3h, i).

After confirming the validity of our TSS detection method, we applied it to detect the lowly abundant RNAs expressed during development. In order to compare developmental stages, we performed MAPCap on RNA isolated from the brains of male and female D. melanogaster third instar (L3) larvae. Previous analysis has shown that Drosophila brain is highly sexually dimorphic[47] and this dimorphism affects sex-specific behaviour, such as male courtship behaviour[48]. Therefore, our MAPCap data of promoter activity could serve as a useful resource to understand these functions. We built an analysis workflow that performs de-multiplexing, mapping, PCR de-duplication, TSS detection and annotation of TSSs (Supplementary Fig. 4a) and processed the MAPCap data through this workflow. TSSs were

called using a strict fold-change threshold of four-fold over the background, followed by a comprehensive functional annotation (see "Methods"). While most of the detected TSSs (70–73%) originated from previously annotated TSSs or 5'-UTRs, 11–12% of the TSSs detected in larvae appeared to originate from enhancers or intergenic regions (Supplementary Fig. 4g). In order to further compare the sex- and stage-specific dynamics of enhancer RNAs, we applied our algorithm to the CAGE data of the adult male and female fly heads, obtained from the modENCODE project (see "Methods"). Combining this data with our MAPCap data, we detected 2793 enhancer RNAs (defined as union of TSS detected within in vivo validated enhancer regions[49]). Interestingly, 1915 (68.6%) of these eRNAs were detected only in one stage, while only 508 (18.2%) were common between all 3 stages (Fig. 2g), indicating a high variability in enhancer activity during development[50]. Further comparison of the stage-specific eRNAs with those detected in all 3 stages revealed that stage-specific eRNAs have significantly sharper TSS ($P = 7.81e-200$, Wilcoxon test, Supplementary Fig. 4b) than the eRNAs commonly detected in all stages. Stage-specific eRNAs also tend to be located significantly away from the nearby annotated TSS, while common eRNAs mostly overlap with an annotated TSS (Supplementary Fig. 4c), suggesting that the common eRNAs either originated from housekeeping enhancers or from a canonical TSS. We performed de novo motif analysis of stage-specific eRNAs in order to find transcription factors important for development. eRNAs expressed in the embryos were enriched for binding site of Foxk1 (regulator of myogenic progenitor proliferation[51]), while larvae-specific eRNAs were enriched for Obp3 (Odorant-binding protein[52]) and Ascl1 (regulator of neuronal development[53]) (Supplementary Fig. 4d). Motifs enriched on eRNAs in the larvae brains were also enriched on eRNAs in adult heads, suggesting that tissue-specific regulation of eRNAs by these factors is established early in development. Further comparing TSS in male and female larvae brains and adult heads revealed 485 potential sex-specific eRNAs. Three hundred and eighty-three (78.97%) of them were female specific (Fig. 2h). These sex-specific eRNAs were not biased to any specific chromosome (Supplementary Fig. 4e). Taken together, our analysis of eRNAs captured regulatory interactions during development.

**Differential expression and promoter usage in MLE mutants.** Our replicate-based experimental design, the ability to remove PCR bias, along with the use of capped ERCC controls provides us with the opportunity to quantify promoter activity between conditions better than standard promoter-profiling approaches. This is highly desirable in the study of dosage compensation as it allows us to compare expression fold-changes across promoters between chromosomes and sexes. In Drosophila, the overall gene expression as well as RNA Pol-II levels have been compared between X and autosomes in cultured cells[30,31] and flies[54]. However an in vivo high-resolution analysis of dosage compensation at promoters has not been done before. Therefore, we performed MAPCap on RNA isolated from the brains of MLE RNA helicase mutant (also referred as knockouts (KOs)) D. Melanogaster third instar (L3) larvae, a stage where male lethality manifests as a result of defective dosage compensation[55]. We then compared the mutants with wild-type (control) larvae with the aim to quantify how different promoters respond to the resulting defects in dosage compensation (Fig. 3a, b, Supplementary Fig. 4f, g). Furthermore, this allowed us to study in vivo MLE sensitivity of autosomal promoters, which has not been studied previously in detail. We also performed a ribo-depleted RNA-seq

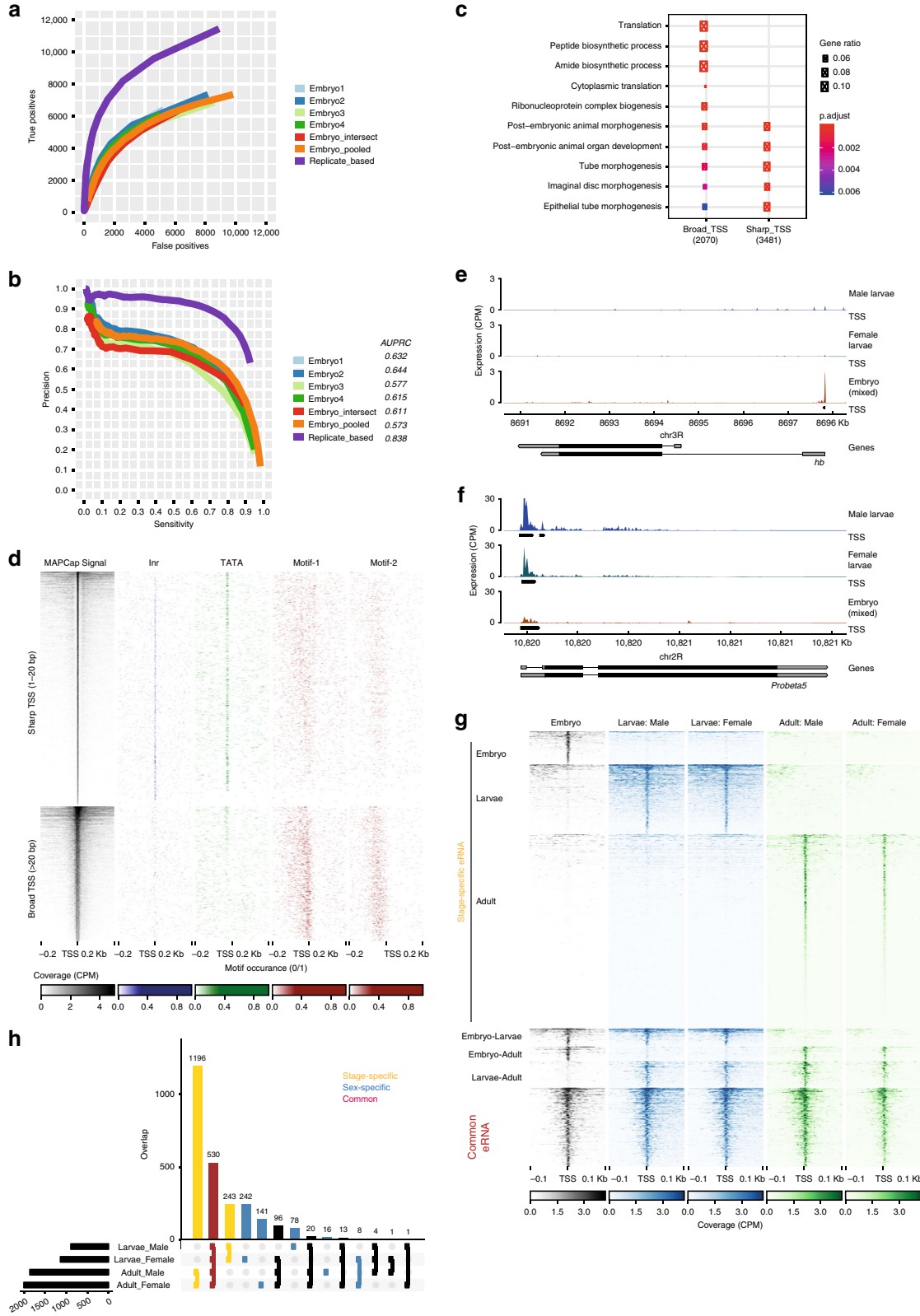

independently using the same set-up, in order to check the concordance between the two protocols. Gene-level expression estimates obtained from both protocols show high correlation (Pearson's $R = 0.85$, Fig. 3c, Supplementary Fig. 4h, see "Methods"), motivating us to perform differential expression analysis using MAPCap.

We first asked whether the MAPCap experiment using biological replicates can be used to obtain differential expression statistics, similar to RNA-seq. We computed gene-level differential expression estimates between wild-type male and female brains on the 5'-UTRs of genes, in order to exclude expression bias due to gene length and splicing (see "Methods"). The

**Fig. 2** Replicate-based analysis improves accuracy of transcription start site (TSS) detection. **a** Evaluation of true and false positives on MAPCap (Multiplexed Affinity Purification of Capped RNA) data of embryos. TSS was detected by paraclu using individual replicates (embryos 1–4), pooled replicates (embryo_pooled), or individual replicates followed by intersection (embryo_intersect) and compared with the "local enrichment" method (replicate based) that uses all four samples. The detection accuracy improves using this method. **b** Precision-recall curve (PRC) of subsampled MAPCap data, comparing paraclu (embryos 1–4, pooled and intersect) with replicate-based method (AUPRC = area under PRC). **c** Comparative Gene Ontology enrichment of "sharp" (upto 20 bp) and "broad" (>20 bp) TSSs detected by the local enrichment method. **d** MAPCap signal (counts per million), as well as motif presence (detected by FIMO) for the "sharp" and "broad" TSS. **e, f** Genome snapshot of MAPCap signal and the detected TSS using the "local enrichment" method on a developmental gene (hunchback, hb) and a housekeeping gene (proteosomal subunit, Prosbeta5). **g** TSS detected on validated enhancer regions (eRNAs) in embryo, L3 larvae brains (MAPCap), and adult heads (modENCODE CAGE). **h** Overlap of eRNAs detected between male and female larvae brains and adult heads show that most eRNAs are stage-specific (yellow) rather than sex-specific (blue)

differential expression statistics from both protocols show a high concordance (Fig. 3d, false discovery rate (FDR) Up: 9.47e−28; Down: 2.8e−09; see "Methods"). Although RNA-seq predicted more differentially expressed (DE) genes at a fixed FDR cutoff, the direction of changes in MAPCap were in agreement with RNA-seq (Supplementary Fig. 5a,c). The GO term analysis of genes DE in MAPCap and RNA-seq also point to the same biological processes (Supplementary Fig. 5b), suggesting that MAPCap produces results in concordance with RNA-seq.

In order to detect changes in the level of TSS expression, we performed differential expression analysis at all TSSs detected by MAPCap in wild-type and mutant larvae. We utilized counts from our capped spike-in RNAs for normalization of fold-changes (see "Methods"). While most TSSs did not show a strong significant sex-specific (or sexually dimorphic) expression bias (Supplementary Fig. 5d), both MAPCap and RNA-seq identified a high expression bias for *Msl2* in males. We also detected sex-specific activity of promoters such as male-specific activity of *Lsp1alpha* and *beta* and female-biased activity of *Sp1* promoter (Supplementary Fig. 5a,d, see "Discussion"). The analysis of wild-type male and female brains indicated male-specific usage of three *roX1* promoters, along with one *roX2* and *Msl2* promoter. Interestingly, we observed a developmental stage- and sex-specific promoter usage of *roX1* RNA (Fig. 3e, see "Discussion"). Comparison of wild-type with mutant larvae revealed a global decrease in promoter activity of most expressed genes on the male X chromosome (Up: 241, Down: 1782; at FDR < 0.05, Fig. 3f), while females showed almost no significant effect (Up: 0, Down: 4; at FDR < 0.05). We also tried alternative normalization methods that do not require spike-ins, such as TMM[56], upper quartile[57] and RLE/DESeq2[58]. These methods, however, produced a more balanced number of upregulated and down-regulated TSSs. Among all methods, TMM showed the highest bias towards the proportion of downregulated genes on X chromosome, however, producing the smallest set of DE TSS (Up: 271, Down: 260; at FDR < 0.05; Supplementary Fig. 5i). Similar result was obtained for gene-level differential expression analysis in RNA-seq with DESeq2 (Supplementary Fig. 6a), suggesting that spike-in normalization provides more useful differential expression estimates.

Although a small number of TSSs showed sexually dimorphic expression bias (25 female and 29 male biased TSSs at FDR < 0.1), our visual inspection of these loci revealed an interesting effect: these sexually dimorphic TSSs seem to be downregulated in MLE mutants (Supplementary Fig. 5g, h). Indeed, statistical analysis showed a clear trend: promoters with sex-specific activity were downregulated in corresponding MLE mutants, for both males (FDR = 4.04e−13) and females (FDR = 7.21e−11, Fig. 3f, Supplementary Fig. 5e, f). While 31% (9 out of 29) of these male-specific TSSs were on the X chromosome, only 16% (4 out of 25) of female-specific TSSs were on the X. Taken together, this analysis indicated a potentially unexpected function of MLE in the regulation of sexually dimorphic promoter activity on both X and autosomes.

**Quantifying changes on X chromosome during dosage compensation defects.** We next focussed on the effect of MLE on male X chromosome. Of the 1782 TSSs significantly down-regulated in MLE mutant males, 605 (34%) were on X chromosome. On the contrary, only 14 (6%) of the upregulated TSS were from X chromosome. X-chromosomal promoters showed a median 1.92-fold downregulation in male mutants (median log2 fold-change = −0.9436, Median Std. Err = 0.472). Divided by expression quartiles, the TSSs showed a downregulation of 1.86–2.02-fold, with the highest variability observed in TSSs with the least expression (Supplementary Fig. 6b). Further, we found that the TSSs on X are on average (median) downregulated by 1.38-fold compared to autosomes in males (Figs. 4a) and 1.8-fold compared to X chromosome in females. We then examined the effect on TSSs based on their location and biotype. Annotated start sites and UTRs respond the strongest in the KOs, while intergenic and antisense TSSs responded the least (Supplementary Fig. 6c). Comparing the biotypes of TSSs, we found that long non-coding RNAs (lncRNAs; which include the strongest MLE targets, *roX1* and roX2) respond the strongest to the MLE KO. Interestingly, we find that TSS-distal enhancer RNAs show a downregulation remarkably similar to the protein-coding transcripts (median fold-change: eRNA = 0.488, protein-coding 0.499; P = 0.83, Wilcoxon test, Fig. 4b), suggesting that eRNAs are equally dosage compensated as protein-coding genes.

We next asked which factors are associated with the change in TSS expression in MLE mutant males. When we compared the expression of MLE sensitive (FDR < 0.05) and insensitive (FDR > 0.5) promoters on the X chromosome, we found that MLE-sensitive promoters have significantly higher expression than the MLE-insensitive promoters (Fig. 4c, P = 1.82e−06; two-sided T test). Promoters with very low expression could also show no significant change due to the low power of differential expression analysis[58]. We therefore restricted the comparison between these groups to a subset of promoters with normalized counts between 100 and 100,000, which have similar mean expression between the two groups (P = 0.1644, two-sided T test). We first tested how the genes with MLE-sensitive and -insensitive promoters compare to a previously published study that assigned a dosage compensation score to transcripts based on their expression during the onset of dosage compensation in early embryonic development[28]. Genes with MLE-sensitive promoters seem to be tightly dosage compensated (median compensation score = 0.998), while genes with MLE-insensitive promoters show a compensation score similar to those on autosomes (Supplementary Fig. 6d). Further comparing these two groups, we found that the MLE-sensitive promoters were located significantly closer to the HAS, which are considered to be sites important for MSL complex recruitment on the X chromosome (Fig. 4d, P = 1.07e−08; two-sided T test). Furthermore, MLE-sensitive promoters were found to be significantly closer to the boundaries of TADs in flies (Fig. 4e, P = 2.12e−09; two-sided T test). Overall, this analysis suggested that promoter location plays an important role in its sensitivity to dosage compensation.

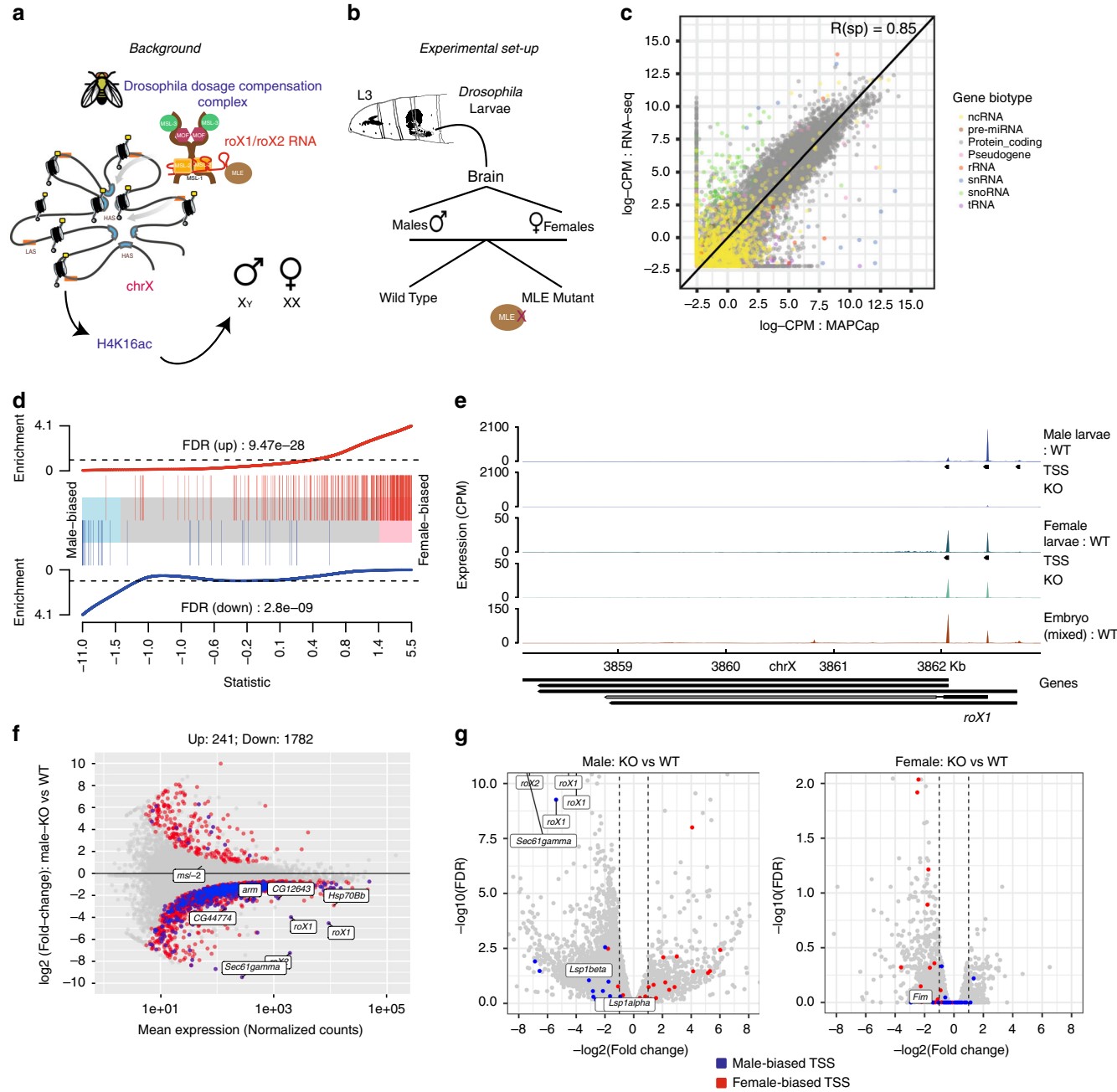

**Fig. 3** Analysis of differential promoter activity in the male and female larvae brains. **a** Background: The dosage compensation complex (DCC) contains an RNA helicase MLE (maleless) that incorporates roX RNAs into the complex. DCC first targets certain high-affinity sites (HAS) on the X chromosome, followed by the spread to the low affinity sites (LAS). This spread is guided by three-dimensional conformation of the X chromosome and requires the recognition of *roX* RNAs by MLE. **b** Experimental set-up: We extracted brains from L3 larvae of flies that are mutant (leading to Knock-out) of MLE and compared them to wild-type flies. **c** Comparison of gene-level expression estimates (log-CPM) obtained from MAPCap (Multiplexed Affinity Purification of Capped RNA) and ribo-depleted RNA-seq in wild-type males (see Supplementary Fig. 3c for females). Expression of most genes are similar except for small nucleolar RNAs and small nuclear RNAs (exclusively depleted in MAPCap). **d** Barcode plot of RNA-seq differential expression results over *t* statistics obtained from MAPCap. The vertical red and blue bars mark differentially expressed (DE) gene in RNA-seq at false discovery rate (FDR) < 0.05 and the lines above and below the bars represent their enrichment in MAPCap data. **e** Promoter usage of roX1 between sexes and stages. **f** MA plot of DE TSSs obtained from MAPCap after spike-in normalization. X chromosome promoters are marked in blue while autosomes are marked in red. **g** Volcano plot of differential expression estimates on TSSs from MAPCap data in female (top) and male (bottom) MLE mutants. TSS with female-specific activity (at FDR < 0.1) are downregulated in female mutants (marked in red) while TSSs with male-specific activity (marked in blue) are downregulated in male mutants

**icetea simplifies TSS detection and expression analysis from promoter-profiling data.** We implemented the processing and analysis methods described in this manuscript in an easy-to-use R package called icetea. icetea can perform sample de-multiplexing and removal of PCR duplicates for MAPCap and RAMPAGE protocols. It also employs our new TSS detection approach that takes advantage of biological replicates (Fig. 4f). Further functions for quality control and quick annotation of detected TSS are also implemented (Supplementary Fig. 6e, f, see "Methods"). Differential TSS expression analysis can be performed between groups

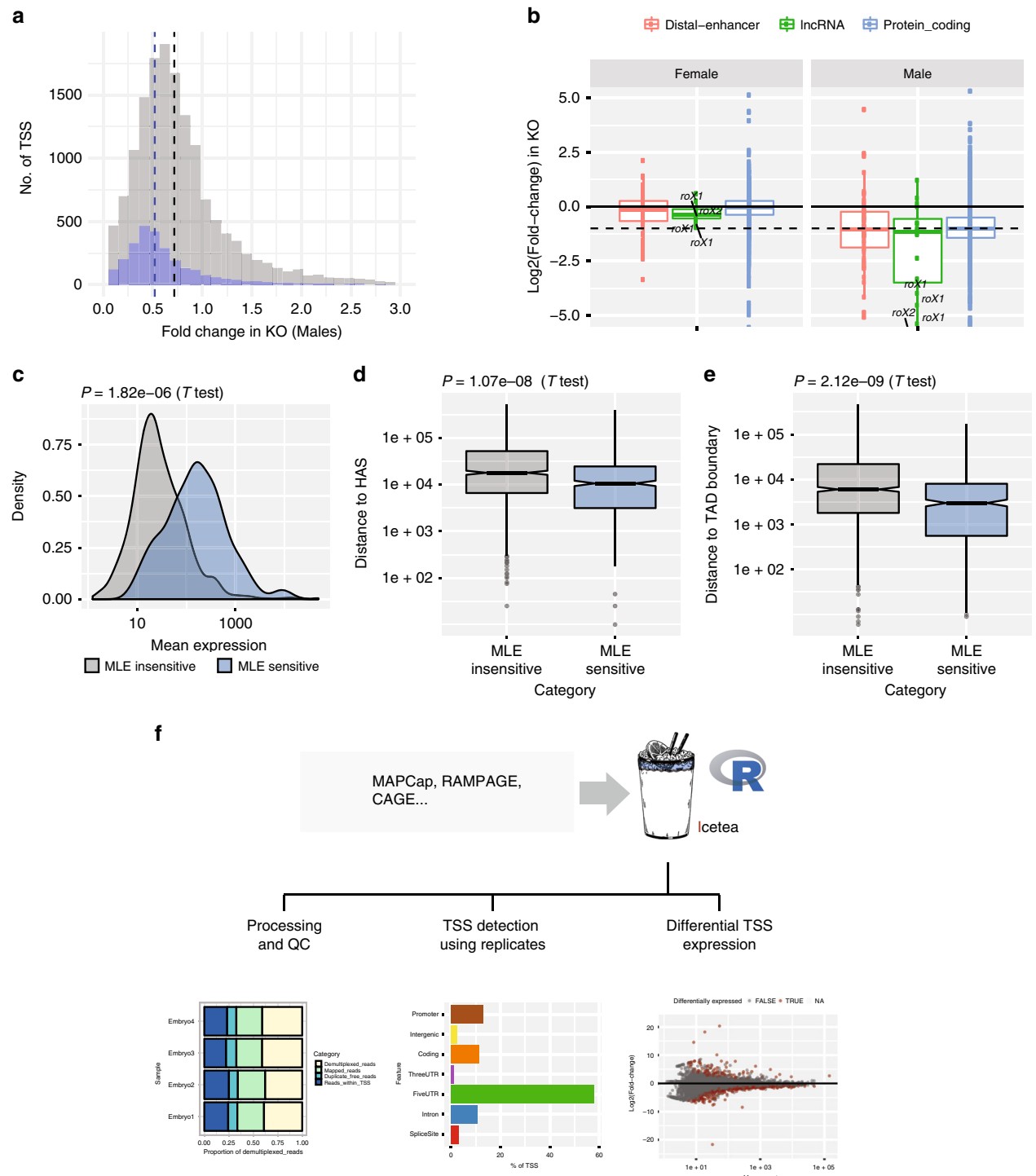

**Fig. 4** MLE (maleless) sensitivity of X chromosome promoters. **a** Histogram of fold-changes in transcription start site activity in male knockouts (KOs) compared to wild type, for genes on X (blue) and autosomes (grey). **b** Effect of KO on the expression of eRNAs ($n = 63$), lncRNAs ($n = 16$) and protein-coding genes ($n = 1721$) on the X chromosome (boxplots = median and IQR, dashed line = 2-fold). **c** Mean expression (replicates) of MLE sensitive (605) and insensitive (1047) X-chromosomal promoters. **d** Distance to high-affinity sites (HAS) for a subset of MLE sensitive (377) and insensitive (136) X-chromosomal promoters (in bp). The subset was selected such that mean expression difference between the two groups become insignificant. **e** Distance to topological domain (TAD) boundary for the same subset of promoters (in bp). **f** A summary of the functionalities of the "icetea" bioconductor package. Outputs of three functions: plotReadStats, annotateTSS, and detectDiffTSS, are shown (left to right)

of samples using either previously established internal normalization methods (see "Methods") or external/spike-in normalization, allowing accurate quantification of changes in transcript expression. icetea is especially suitable for end-to-end analysis of paired-end 5' profiling techniques, such as MAPCap and RAMPAGE. However, it can easily be used for analysis of CAGE, GRO-Cap and other promoter-profiling protocols. icetea is open source and available for use via Bioconductor (https://bioconductor.org/packages/icetea) and the source code is available on GitHub (https://github.com/vivekbhr/icetea).

## Discussion

In this study, we introduce MAPCap, an easy to perform promoter-profiling technique that allows processing dozens of samples in about 16 h of time. We further combine MAPCap with the icetea software that simplifies TSS detection and analysis of differential TSS activity using biological replicates in about 7–8 steps. The MAPCap protocol along with icetea analysis provides most of the benefits of CAGE and RNA-seq but at a fraction of total cost and time of performing both the protocols. We demonstrate that MAPCap shows sufficient enrichment at TSS for starting material up to 100–500 ng. This is >10-fold lower than the latest CAGE protocols[9,59], 5–10-fold lower than RAM-PAGE[37] and in the same range as nanoCAGE[60]. Two recent protocols, SLIC-CAGE[7] and C1-CAGE[8], have shown ≤10-fold input requirements than MAPCap, but they require more PCR cycles, together with specialized reagents and equipments. In this comparison, we multiplexed low-input samples with those with higher RNA abundance (5 μg), which might affect TSS enrichment of these samples due to RNA composition bias[56]. Therefore, we expect that multiplexing samples with similar RNA abundance, along with a replicate-based analysis implemented in icetea, could further improve TSS detection and differential analysis on low-input samples. Instead of cap-trapping (nAnT-iCAGE) or template-switching (nanoCAGE, RAMPAGE), MAPCap utilized affinity purification using an antibody, eliminating several protocol-specific optimization or difficult steps. Further, the s-oligos used by MAPCap can also be used for in vivo RNA–protein interaction analyses[33] or for RNA-seq, allowing a wider scope of integrative analysis. We propose that this approach would prove optimal for transcript annotation and gene expression analysis of newly assembled as well as annotated genomes.

Analysis of MAPCap data revealed the dynamics of sex-specific promoter usage as well as transcript expression. Comparison of promoter activity between wild-type males and females showed male-specific expression of multiple subunits of *Larval serum protein 1* (*Lsp1*) on the X chromosome (Supplementary Fig. 5d). *Lsp1* and 2 proteins are highly expressed in fat bodies, which are commonly found associated with brain and are important for metamorphosis and reproduction[61]. *Lsp1alpha* is a recently evolved gene that has been shown to lack dosage compensation on its endogenous site the X chromosome, although it is capable of dosage compensation at an ectopic location[62,63]. Confirming this observation, we find *Lsp1alpha* as one of the least MLE-sensitive promoter in our analysis (FDR = 0.91). Interestingly, the *beta* and *gamma* subunit of the *Lsp1* heterohexamer complex, which are located on chromosomes 2 and 3, respectively, also show male-specific expression, although they do seem to show some level of MLE sensitivity (FDR = 0.08 and 0.14, respectively). Similarly, we detect other sex-specific expression of genes involved in mating behaviour and reproduction, such as male-specific *Obp99b* and female-biased expression of *Sp1*[64], *peb*[65] and other genes (Supplementary Fig. 5a, d). Interestingly, many of these sex-biased genes, even those on autosomes, show MLE sensitivity in females. Unlike males, MLE is seen to bind on all chromosomes in females[66]; however, the function of MLE on female autosomes is less well studied. Our observations point towards a role of MLE in regulating the expression of sex-specific genes in both males and females, independent of X chromosome.

We also observed stage- and sex-specific expression of *roX1* isoforms through utilization of different promoters (Fig. 3e). *roX1* RNA, which becomes sexually dimorphic during development[67], is predicted to produce five transcript isoforms using three different promoters. While embryos seem to utilize the 3′-most promoter, female larvae (with a very low *roX1* expression) tend to

use the middle and 3′-promoter equally. Male larvae, which clearly show a very high expression of *roX1*, mostly utilize the middle promoter. As this promoter variability leads to a shorter isoform in male larvae, this points to a developmental stage-specific role of *roX1* isoforms in dosage compensation. All three promoters display a loss of expression upon MLE KO, suggesting that MLE is required for the stable expression of all three *roX1* RNA isoforms[67,68].

After appropriate external spike-in normalization, TSSs on the male X chromosome show a median 1.92 downregulation upon MLE KO. Further comparison between chromosome and sexes corroborates previous analysis of dosage compensation, which suggested a median 1.1–1.8-fold dosage compensation between X and autosomes[69] and a 1.6-fold dosage compensation between sexes[70]. It should be noted that these analyses are based on hemizygosity/copy number changes between genes while ours is based on a direct perturbation of the dosage compensation system. Promoter-proximal MAPCap analysis clearly indicates significant changes in TSSs upon depletion of MLE. It will be interesting to use methods such as NET-seq[71] and TT-seq[72] in order to distinguish between the contribution of initiation and elongation rates in dosage compensation.

Further, our analysis indicated factors that influence variability in dosage compensation at promoters. Previous analyses from our group[73] and others[74] have shown that X chromosomal HAS for the MSL complex are located close to TAD boundaries and form a cluster of spatial interaction on X chromosome. Using our data, we investigated whether the distance to the HAS or TAD boundaries could have a functional consequence in promoter activity during dosage compensation. We find that certain promoters have a low sensitivity to MLE, indicating that transcripts originating from these loci might "escape" dosage compensation in males. These MLE-insensitive promoters have lower wild-type expression compared to the MLE-sensitive promoters and are located further from HAS as well as TAD boundaries. This sounds counterintuitive since at least a subset of HAS have been shown previously to support a recruitment of MSL complex independent of MLE[75–77] and therefore promoters close to HAS can be hypothesized to dosage compensate without MLE. However, despite studies suggesting a weak recruitment, the functional activity of the MSL complex at these promoters is uncertain. Our analysis does not resolve the issue whether MSLs could be recruited to HAS without MLE, but it does suggest that these promoters fail to dosage compensate in the absence of MLE. Interestingly, a recent study of *roX1/2* double mutants (which produce an effect largely similar to MLE mutants) has performed a similar analysis and arrived at a similar conclusion that genes >30 kb away from HAS are less sensitive to the absence of *roX* and to the bound MSL complex[78]. Furthermore, our observations on the relationship between MLE sensitivity and TAD boundaries seems to be in line with observations on X inactivation in mammals, where genes shown to escape inactivation are shown to be near TAD boundaries[79].

By combining our promoter-profiling protocol MAPCap and analysis software icetea, the data sets and analyses presented in our study serve as a useful resource on sex-specificity and MLE sensitivity of *Drosophila* promoters. As our approach is annotation-agnostic, it could easily be applied in future to genomes of other species in order to understand genome-wide differences in promoter activity between tissues, sexes and conditions. Further, the ability to de novo detect and quantify differential expression of TSSs would prove useful for studies that require detection and relative quantification of lowly expressed RNAs, such as tissue-specific changes in eRNA expression, effects of genomic imprinting or changes in lncRNA expression between wild-type and mutants/KOs. Finally, we envision that our fast and

easy-to-use approach would be suitable for clinical investigation of changes in non-coding RNA expression in diseases.

## Methods

**Cells.** S2 cells (gift from the Butros Laboratory, Heidelberg) were cultured in Express Five SFM media (Thermo Fisher) supplemented with 10% (v/v) Glutamax (Thermo Fisher). Cultures were maintained adherent or in shaking incubators at 27 °C at a speed of 80 rpm. Cells were kept at a density of 1–16 million/mL.

**Fly culture and genetics.** Fruit flies (*D. melanogaster*) were reared on standard cornflour-molasses medium [1 L water, 12 g agar–agar threads, 18 g bakery yeast, 10 g soya flour, 80 g cornflour, 22 g molasses, 80 g malt extract, 2.4 g 4-hydroxibenzoic acid methylester (Nipagin), 6.25 mL propionic acid] at 25 °C, 70% relative humidity and 12 h dark/12 h light cycle. mle9 is a gamma ray-induced loss-of-function allele (deletion) of mle, generated by Scott M and Lucchesi J, and first reported in refs. [32,80]. mle9, cn, bw/CyO flies (BDSC #5873) have been ordered from the Bloomington *Drosophila* Stock Center (BDSC) and rebalanced with the help of w; CyO, Act5C-GFP/If to generate the w; mle9, cn, bw/CyO, Act5C-GFP flies used in this study.

**Generation of capped ERCC spikes.** ERCC mix 1 was reverse transcribed using SuperScript III (Invitrogen). Ten spike sequences were chosen from the ERCC spike DNA and PCR primers were designed to produce ~500-bp long DNA fragments (Supplementary Information). At the 5′-end, we inserted a T7 class II promoter $\phi$ 2.5, which has been shown to create more homogenous 5′-end transcription promoter sequences[81]. Spikes were in vitro transcribed using the T7-FLASHScribe Transcription Kit (CellScript) according to the manufacturer's instructions and purified using the MegaClear Kit. In vitro capping was performed with Vaccinia Capping System (NEB). Potentially uncapped RNAs were degraded by treatment of spikes with Polyphosphatase (Epicentre) and Terminator exonuclease (Epicentre). Samples were cleaned using Oligo Clean and Concentrator (OCC, Zymo Research) and concentrations were measured on Qubit. A master mix was created where each subsequent spike was added at half the concentration of the previous spike, starting from 8 fmol/μL.

**MAPCap library preparation.** For detailed protocol, please refer to Bhardwaj et al. (Nature Protocol Exchange, https://doi.org/10.21203/rs.2.9396/v1). Briefly, RNA from S2 cells, embryos (stage 15, 50 embryos per replicate) and dissected brains from third instar larvae (10 brains per replicate) was extracted using the DirectZol Kit (Zymo Research). RNA was eluted in 25 μL of RNase-free water. The concentrations were adjusted and capped ERCC spikes were added at 0.05% of input amount. For our stage 15 embryos, we diluted each replicate to 5 μg of total RNA and added the spike-ins ranging from 0.0004% to 0.05%. To remove abundant capped RNAs (snRNAs, snoRNAs) as well as rRNA contamination, we added antisense DNA oligos (Supplementary Information) targeting the RNA species detected from a preliminary MAPCap run. Eight-μL of oligo mix were added together with 4 μL of 10× terminator buffer A (Epicentre). The RNA was heated to 70 °C for 2 min followed by an active cooling in the Thermomixer (Eppendorf) to 37 °C. Upon reaching this temperature, 1 μL of RNaseH (Life/Invitrogen) was added and incubated at 37 °C for 30 min. The samples were then heated to 70 °C for 2 min, put immediately on ice for 1 min and 1 μL of Terminator exonuclease was added for 1 h at 30 °C. RNA was purified using RNA clean and concentrator (Zymo Research) and eluted with 100 μL TE buffer. The samples were fragmented using a Covaris E220 Ultrasonicator (200 cycles/burst, Duty cycle 5, 175 W, 10%) for 180 s per sample. Fragmented RNA was incubated with 2.5–5 μg of anti-m7G antibody (SYSY, cat. no. 201001) pre-coupled to Protein G magnetic beads for 1 h in IPP buffer (50 mM Tris-HCl pH 7.4, 150 mM NaCl, 0.1% NP-40) rotating at 4 °C. Beads were washed three times with IPP and RNA 3′-ends were dephosphorylated using PNK for 30 min at 37 °C. Beads were washed and the s-oligo was ligated using T4 RNA Ligase 1 for 1 h at 25 °C. s-oligos contain barcode and random nucleotides in the following pattern NNNNNTTTTTTNN (N = random nucleotide; T = barcode nucleotide). Excess s-oligos were washed away with IPP buffer and samples were pooled together. After 30 min of treatment with rSAP at 30 °C to dephosphorylate the s-oligo, the RNA was released from the beads using Proteinase K treatment and column purification (Oligo Clean and Concentrator (OCC), Zymo Research). Isolated RNA was reverse-transcribed using SuperScript III (Invitrogen) for 10 consecutive minutes at 42, 50, 55 and 65 °C. After 30 min treatment with RNaseH, the cDNA was column-purified using OCC and circularized with CircLigase2 for 2–16 h. One-μL of circularized cDNA was taken to determine the amplifications cycles using quantitative PCR. After PCR amplification, the libraries were cleaned up twice using 1× Ampure beads (Beckman Coulter), quantified with Qubit (Thermo Fisher Scientific) and the quality was assessed on Bioanalyzer (Agilent). MAPCap libraries were sequenced on Illumina NextSeq 500, 3000 or HiSeq 2000.

**MAPCap in mouse ESCs.** The hybrid mouse female ESC line F1–21.6 (129Sv-Cast/EiJ) was grown on Attachment Factor (Gibco) coated dishes in 2i medium containing Dulbecco's modified Eagle's medium (Gibco) supplemented with 15% KnockOut Serum Replacement (Gibco), Penicillin/Streptomycin (Sigma),

1000 U/mL LIF (Millipore), sodium pyruvate (Gibco), non-essential amino acids (Gibco), Glutamax (Gibco), 100 μM β-mercaptoethanol (Gibco), 5 μg/mL insulin (Sigma), 1 μM PD0325901 (Axon) and 3 μM CHIR99021 (Axon). Cells were maintained in a humidified incubator at 37 °C and 5% $CO_2$.

Total RNA was prepared from 70–80% confluent 10-cm dishes using Trizol (Ambion) according to the manufacturer's instructions. Poly(A)-RNA was isolated from 5 μg of total RNA (mixed with spike-ins added at 0.05% of input) using Dynabeads Oligo-(dT)25 (Thermo Fisher). In order to remove traces of rRNA contamination in poly(A)-RNA samples, anti-sense DNA oligos were used with a sequential RNaseH and terminator exonuclease treatment. RNA fragmentation, IP and library preparation were performed as described above.

**Processing of MAPCap data.** Paired-end FASTQ files were trimmed for adaptors using Trimmomatic[82] (v 0.3.7). Samples were de-multiplexed by icetea (v0.99, demultiplexFastq) using provided barcode information and mapped to the dm6 genome using Rsubread[83] (v 1.22.3, mapping wrapper provided in icetea). For de-duplication, we consider all reads mapping to the same 5′-position and having the same random barcode as duplicates and only keep the first instance of each such alignment (using icetea—filterDuplicates). BigWigs were created using deepTools[84] (v3.0.2) bamCoverage and bamCompare, with the option "–offSet 1–binSize 1–normalizeUsing CPM". Quality control was performed using deepTools and multiQC[85] (v1.3). Genomic regions were plotted using pyGenomeTracks[86] (v2.0). The MAPCap data processing workflow (described in Supplementary Fig. 3A) is available at https://github.com/vivekbhr/cage_pipeline. For allele-specific mapping of mouse ESC data, we mapped the MAPCap data on a modified GRCm38 (mm10) genome using STAR (v2.6.1)[87] with options "–sjdbOverhang 100–readFilesCommand zcat–outSAMunmapped Within–outSAMtype BAM SortedByCoordinate–outSAMattributes NH HI NM MD". The genome was created using SNPsplit[88] using options "–dual_hybrid–full_sequence" and strains 129S1_SvImJ and CAST_EiJ using the VCF files (v5) from the mouse reference genome project[89]. After mapping, the alleles were sorted using SNPsplit sort with option "–paired". TSS were detected using the paraclu method[42] on sorted paternal and paternal genomes using parameters described in section "Evaluation of TSS accuracy".

**Processing of external data sets.** For comparison with MAPCap, external data sets were downloaded from GEO (Supplementary Table 1). We trimmed the first "G" nucleotide using trimgalore (v0.4.4) for CAGE, with parameters: trim_galore–clip_R1 1–trim-n; and for RAMPAGE, with parameters:–paired–clip_R1 4–trim-n. Data were mapped to the dm6 genome using subread (subjunc; parameter: -d 10; same as with MAPCap). PCR duplicates were removed in CAGE data using picard MarkDuplicates (v2.13.2), while for RAMPAGE data using icetea (filterDuplicates) considering the 15 bp sequence from read 1 as the random barcode. To calculate correlation of signal, we counted reads on all non-overlapping 5′-UTRs of genes (22,148 regions from 13,378 genes) using the bioconductor package GenomicAlignments (summarizeOverlaps, mode = "IntersectionNotEmpty"). Fragments were counted for all data sets instead of the single-end CAGE data. Counts were normalized using counts per million (CPM) for correlation and plotting. To additionally compare correlation between replicates at the detected TSS, we downsampled MAPCap (embryo) and CAGE (S2 cells) data to 1 million reads and performed peak calling using paraclu on pooled reads before or after duplicate removal. We then computed Pearson correlations and plotted the signal (log(CPM + 1)) between replicates on these detected TSSs.

For evaluation of TSS, we processed the DNAse-seq data of late-stage embryos[43] and S2 cells[90] using snakePipes (v1.1.2) DNA-mapping workflow[91] with the parameters "–trim–fastqc–mapq 5 dm6". We then performed peak calling using MACS2 (v 2.1.2)[44] using parameters "-f BAM -g 142573017 -q 0.001". Replicates were merged for S2 cells before peak calling. We also processed the ChIP-seq data for three active histone marks (H3K4me3, H3K4me2 and H3K27ac) from modENCODE[92] and processed them using snakePipes DNA-mapping (parameters: "–trim–fastqc–mapq 5–dedup dm6") and ChIP-seq (parameters: "–single-end", MACS2 -q "0.005") workflows. The union of resulting peaks were considered "active peaks" for the analyses.

For comparison of MAPCap and nAnTiCAGE in mouse ESCs[7], we mapped both data sets using STAR (v2.6.1)[87] on the GRCm38 (mm10) mouse genome. We downsampled the mapped files to 1 Mil and the duplicate-removed file (for MAPCap) to 100K reads. We performed the quality checks, correlation of signal and metagene profiles on all annotated genes (Ensembl-91) using deepTools (v3.1).

**Evaluation of TSS accuracy.** For evaluation of TSS detection accuracy, we used the paraclu method[42] to cluster CAGE tags from CAGE, RAMPAGE and MAPCap data. We used the 12–14 h sample from modENCODE, and merged the 12, 13 and 14 h samples from RAMPAGE to compare with merged (embryos 1–4) samples from MAPCap data. All samples were then downsampled to 5 million reads (keeping only R1 for MAPCap and RAMPAGE) and paraclu was run with the parameters: min_value = 1, min_density_rise = 1, min_pos_with_data = 1, min_sum = 1, min_width = 3, max_width = 300 (i.e. all criteria such as minimum reads used for clustering and minimum density of reads per cluster etc. were kept to the lowest, and tag clusters of length 3–300 bp were considered for analysis).

The score (sum of reads at detected TSS) provided by paraclu for each tag cluster was used to calculate precision and sensitivity. Scores between 10 and 1000 were plotted for the evaluation of true and false positive (TP/FP) and precision-recall curve (PRC). For the calculation of true and false positives, we used the RNA-seq data of 12–14 h embryo from modENCODE[93] and calculated transcript-level TPMs using Salmon (v0.9.1)[94]. Transcripts with TPM > 1.0 were considered "expressed" and the expressed transcripts that overlapped with a DHS peak but were not detected by the promoter-profiling methods were considered "false negatives". TSSs that were detected by the methods in genic regions but did not overlap with a known TSS (±100 bp) in dm6 (ensembl-79) annotation were considered "false positives". For comparison between MAPCap and other protocols, we obtained optimal F1-score for each protocol independently by testing various density_rise (between 1 and 30) and min_sum (between 1 and 50) cutoffs. To add orthogonal evidence for unannotated TSSs, we overlapped the false positive peaks with DNAse-seq data from late-stage embryos and ChIP-seq "active peaks" (see above). For comparison between icetea and paraclu, we additionally also compared the TP/FP and PRC at various density_rise cutoffs (difference between maximum and minimum signal at TSS). To show application of icetea on data from MAPCap and CAGE at equal depth, we downsampled the MAPCap data (embryos, 4 replicates) and CAGE data (S2 cells, 2 replicates) to 1 million reads and repeated the above analysis.

**TSS detection and differential TSS usage analysis using replicates.** For the detection of TSSs using replicates, we first count the 5'-end of reads in 10 bp sliding windows ($w$) across the genome for all samples (with a slide of 5 bp). For each window, we also calculate all 5'-ends of the reads falling into the corresponding 2 kb background region ($b$) centered at the window. Counting is done in a strand-specific way, using the *intersectionStrict* mode. We then calculate the fold-change (delta) of each window with respect to the background as:

$$\delta = \mathrm{Avg}(\hat{w})/\mathrm{Avg}(\hat{b}) \quad (1)$$

where $\mathrm{Avg}(\hat{w})$ and $\mathrm{Avg}(\hat{b})$ are mean logCPM values across replicates, obtained by a fitting single-group negative binomial (NB) glm implemented in *mglmOnegroup* function of the edgeR package[95]:

$$\hat{Y}_{wi} \sim \mathrm{NB}(M_i P_{wj}, \phi_w) \quad (2)$$

for window $w$ and sample $i$, where $M_i$ is the library size of sample $i$, $\phi_w$ is the dispersion of the window $w$ and $P_{wj}$ is the relative abundance of the 5'-ends in window $w$ for the experimental group $j$ for sample $i$.

For comparison between replicate-based and paraclu method, we ran paraclu on samples using parameters described in the previous section, while for our method we obtained TSSs using a two-fold local background cutoff, followed by merging the nearby enriched windows. After merging, we re-calculated the average (mean) fold-change of the merged windows and took another cutoff of 1.5-fold, in order to obtain the final TSS. The "TSS score" (mean fold-change of merged windows) was used to evaluate the true and false positives for the analysis. Since the range of scores obtained per TSS is very different between paraclu and our method, there is no comparable cutoff for comparison of precision and sensitivity.

For differential TSS usage analysis, strand-specific counting is performed in the same way, on the union of TSSs detected across samples. Library sizes were normalized using the size factors obtained from ERCC counts using median of ratios method from DESeq2. The differential expression analysis was then performed in DESeq2 using "nbinomWaldTest" function. TSSs with adjusted $P < 0.05$ were considered significantly different between tissues and sexes.

To perform differential gene expression analysis from MAPCap data, we summed the counts obtained from all 3'-UTRs of a gene into one and performed the normalization and differential expression using DESeq2. Spike-in normalization was performed the same way as above.

**TSS annotation.** For a comprehensive annotation of our detected TSSs, we first created a mutually exclusive set of annotations from dm6 (ensembl-79) GTF file, by first separating genic from intergenic regions, followed by ranking them in this order (5'-UTR > CDS > 3'-UTR > Introns; and sense > antisense). Further the features were re-annotated by overlapping them with enhancers[49] and repeats (RepBase release 20140131). The annotation pipeline is available as part of the full MAPCap data processing pipeline at https://github.com/vivekbhr/cage_pipeline.

**Promoter width and motif analysis.** To evaluate the promoter width distribution obtained from icetea analysis, we divided our 12,921 detected TSSs into "broad" and "sharp" categories, by taking arbitrary cutoffs: > 20 bp (36.7%) and <20 bp (63.3%), respectively. We performed GO enrichment analysis of the two categories for biological processes (BP) terms and plotted them using the clusterProfiler bioconductor package[96] ($P < 0.01$, $q < 0.05$). Further, we extracted the FASTA sequences associated with the two categories from the dm6 genome using the BSgenome package. We used 1000 randomly selected sequences from each category (without replacement) and extracted a 20, 50 and 100 nucleotide region centred at TSS for sharp TSS, broad TSS and eRNA TSS analysis. De novo motif enrichment analysis was performed via *meme*[97] (v4.11) using the parameters: -dna -nmotifs 10 -minw 3 -maxw 12 -seed 123 -maxsize 10000000. For visualization purpose, we ran

FIMO[98] (v 4.11.1) using the PWMs obtained from our de novo meme predictions on the genome, using the parameters: –max-strand–thresh 1e-3, and converted the resulting output to a bigWig file with binarized scores (1 = presence, 0 = absence). We additionally downloaded the TATA motif from the JASPER database[99] (2018) and created the bigwigs the same way. Plotting was done using deepTools[84] (v3.1) computeMatrix and plotHeatmap functions using bin size of 1 (-bs 1).

**Comparison between RNA-seq and MAPCap.** To perform a fair comparison between MAPCap and RNA-seq, we extracted RNA from an independent set of fly brains using the same set-up as MAPCap. Brains were obtained from female and male L3 larvae, where three pools of larvae were utilized as replicates. Five µg RNA per replicate was used and samples were sequenced using the ribo-depleted Illumina TruSeq protocol at the depth of ~20 million each. The data were then processed via snakePipes RNA-seq pipeline[91] (v1.0.0 beta) using the options "-m alignment,deepTools_qc–star_options–limitBAMsortRAM 60000000000–outBAMsortingBinsN 30 dm6". snakePipes performed alignment using STAR[87] (v2.6.1a), counting of reads on Ensembl GTF (release 79) via featureCounts[100] (v1.6.1) and quality-checks via deepTools[84] (v3.1.2). The gene-level counts obtained from featureCounts were then used for differential expression analysis via DESeq2[58] (v1.20.0).

Most downstream analyses were performed on R (v 3.5.0) and bioconductor[101] (v3.7). For the comparison of MAPCap and RNA-seq signal at genes, we obtained gene-level counts from MAPCap, using featureCounts with same parameters as RNA-seq. We then converted both RNA-seq and MAPCap counts to average log-CPM for each group, using edgeR[95] (v3.22.3), and checked correlation. For comparison of differential expression results, we counted reads on 5'-UTRs of genes using GenomicAlignments bioconductor package (v1.16.0) using function summarizeOverlaps (mode = "IntersectionNotEmpty"). Since non-coding genes do not have a 5'-UTR, this analysis excluded non-coding genes, leaving 13,378 genes (out of 17,403 total genes). We then plotted DE genes from RNA-seq on to the $t$-statistic obtained from MAPCap fold-changes using barcodePlot from limma[102] (v3.36.2) and performed gene-set enrichment tests using camera[103].

**Comparison of MLE sensitivity and genomic features.** All DE promoters upon MLE KO in male larvae at FDR < 0.05 were considered "MLE Sensitive" (605 promoters on X, 417 genes), while all promoters at FDR > 0.5 were considered "MLE insensitive" (1047 promoters on X, 461 genes). Since the mean expression between the two groups is significantly different, we picked promoters within the DESeq2-normalized counts between 100 and 100,000, where the mean expression between the groups are comparable ($P = 0.164$, two-sided $T$ test). This includes 377 MLE-sensitive TSS (295 genes) and 136 MLE-insensitive TSSs (80 genes).

We obtained the HAS using the ChIRP-seq data of *roX* RNA in *D. melanogaster* from GSE69208[104]. Peak calling was done with MACS2 with parameters: callpeak -f BAM–qvalue 0.01 -g 120,000,000, and the top 250 peaks were used as HAS. To obtain the TAD boundaries, we re-processed previously published Hi-C data from Kc167 cells[86,105,106] on to dm6 genome using hicExplorer (v1.8.1)[86]. TAD calling was done with the parameters: –minDepth 20000–maxDepth 50000–step 2000–delta 0.01–maxThreshold 0.27–lookahead 5. Finally, distance of promoters to HAS and TAD boundaries were calculated using GenomicRanges[107] (v1.32.7) in bioconductor.

**TSS analysis methods implemented in the icetea package.** The icetea package developed during this study implements functions for an end-to-end analysis of data from MAPCap as well as other promoter-profiling methods. Icetea analysis begins by creation of an S3 object of class "CapSet", which holds the metadata for each step of the analysis, along with the detected TSS (as a GRangesList object). The CapSet object can be created either using the raw (multiplexed) FASTQ files or directly using mapped and/or filtered BAM files. FASTQ demultiplexing can performed using the protocol-specific barcode positions, and the data can be mapped using a wrapper to the RSubread package. In order to remove PCR duplicates, the filterDuplicates function of the icetea package iterates through genomic positions and matches the random barcode/UMI at each position. In case of reads with same barcode and 5'-position, only the first instance of the read is kept. This function is currently supported for both MAPCap and RAMPAGE data. TSS detection is performed using the local enrichment method (described above) where the sliding window width can be specified by the users for flexibility. Broad TSS are automatically detected by aggregating multiple instances of consecutively enriched windows (user-defined threshold). In order to compare TSS activity between groups of samples, icetea implements various normalization methods, such as TMM (Robinson and Oshlack[56]) and RLE (Love et al.[58]) (size factors calculated from all detected TSS), "window TMM" (TMM size factors calculated using large windows in the genome), upperquartile[57] and "external" (such as size factors calculated using spike-ins, used in this paper).

**Reporting summary.** Further information on research design is available in the Nature Research Reporting Summary linked to this article.

## Data availability

Raw sequencing data sets, normalized bigWigs and RNA-seq differential expression results have been deposited to GEO under the accession number GSE125831. Detected TSS and differential TSS results are available on Zenodo [https://zenodo.org/record/2638160].

## Code availability

Icetea is available open source at https://github.com/vivekbhr/icetea. All the data presented in the manuscript have been processed via the cage analysis pipeline available at https://github.com/vivekbhr/cage_pipeline.

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

## Acknowledgements

The authors acknowledge the Deep-Sequencing Unit at MPI-IE for data production. We thank Kin Chung Lam, Claudia Keller Valsecchi and Bilal Sheikh for critical reading of the manuscript; Ibrahim Ilik for sharing the s-oligo and the FLASH protocol and Plamen Georgiev for help with MLE mutant and wild-type flies. A.A. and T.M. acknowledge funding from the German Science Foundation (CRC992 "Medical Epigenetics"). This study was supported by the German Research Foundation (DFG) under Germany's Excellence Strategy (CIBSS—EXC-2189—Project ID 390939984).

## Author contributions

V.B. performed all the analysis of data with inputs from G.S., developed the icetea package and wrote the manuscript with input from all authors. G.S. developed the MAPCap protocol

with analysis input from V.B. and performed all the *Drosophila* experiments. N.U.E. performed the mammalian MAPCap experiment. G.S. and V.B. conceived the project with input from A.A. T.M. and A.A. supervised V.B. and G.S., respectively, during the project.

## Additional information

**Competing interests:** A.A. is listed as one of the inventors on the patent application for the s-oligo (application no. PCT/EP2016/066876). The other authors declare no competing interests.

