## [Peer Review File · Nature Communications]

Reviewers' comments:

Reviewer #1 (Remarks to the Author):

Overall, Akhtar and colleagues have combined existing methodologies to develop a novel MAPcap method which is improved because it allows quantitative comparisons not possible with prior approaches for quantifying TSS. They have used early multiplexing of samples, removed PCR duplicates by random barcodes and used external spike in controls to allow accurate quantification of TSS expression. They quantified X-chromosome dosage compensation and discovered that MLE has an interesting sex-specific role in the brain. The authors also confirmed prior observations that different roX promoters are developmentally regulated.

There are several important concerns to address:

- 1) The authors measure dosage compensation without looking at elongation which is a caveat that should be mentioned in the text.
- 2) All figures are very small and many are missing axis labels. Please recheck all labels and increase the size of figures.
- 3) I would like to see more details about their computational icetea method. I read their tutorial on Bioconductor but I do not think they provided enough detail in this paper.
- 4) The authors have shown enhancer RNAs that have stage specific expression but they should have determined whether eRNAs also showed sex-specific expression. Were there no sex-specific eRNAs (Fig. 2g)? This should be mentioned and commented on.
- 5) Embryo and sexed larval brains is not the best comparison because embryo is both male and female and brain is a specific tissue while embryo represents a whole organism. It would be much better if larval and adult brain could be compared in males and females separately.
- 6) Does MLE have any role in females or on autosomes? This should be addressed.

Specific comments:

Their materials and methods missed some detailed information such as the stage and number of embryos used.

Fig 1c-e, What normalization of read counts did they use when calculating the correlation between CAGE vs. MAPCap, and compare them to RNA-seq. They didn't show units on X- Y- axes.

Page 9: why brain tissue, why not study in sexed embryos? Why this stage? is this the stage with brain developed?

Page 12 top paragraph: how similar the DEGs are between RNA-seq and MAPCap? how much in percentage? Any Venn diagram? Did they only used DEGs that presented in both method? Any GO annotation/pathway analysis for the DEGs?

Page 13 top paragraph: MLE mutants caused both autosomal gene and X-lined gene to change (Fig 3f), so the calculation of the X:A ratio in mutant males and females may not be appropriate.

It would be better to plot a histogram of the changes on the X and the changes and autosomes and overlay them.

Page 13 bottom paragraph: How do the MSL sensitive vs. insensitive genes that they identified compare to other studies such as Hamada et al. 2004, Lott and Eisen, or Brian Oliver's studies in

adult heads?

Fig 4a. is t-test a correct test to compare the expression distribution? A K-S test is likely more appropriate to define the difference between two distributions.

Page 17: Can the effects of genomic imprinting be studied using MAPCap? Does it have enough coverage to call the allelic-specific expression at TSS?

Reviewer #2 (Remarks to the Author):

Summary

The authors develop MAPCap, a novel method for sequencing capped transcript 5' ends incorporating s-oligo with unique molecular identifiers (UMI) and sample multiplexing. MAPCap is then performed in *Drosophila* embryos and compared to the CAGE and RAMPAGE protocols. The authors also implement a clustering method to identify TSS which is compared to the previous 'paraclu' method, with advantages when sample replicates are available. Sex specific TSS, and transcribed enhancer loci during *Drosophila* development are identified. MAPCap is also performed in MLE mutant *Drosophila* to examine gene dosage. The authors provide their bioinformatic analysis workflow, and an R package (icetea) for the analysis of MAPCap data.

Main Points

1) The introduction states "Quantification of transcript expression from CAGE is hindered by the PCR amplification bias". This is true for some CAGE methods including nanoCAGE and C1-CAGE, however cap-trapping (nAnTi)CAGE* which performed best in the recent comparative review of 5p profiling methods referenced in the introduction uses no PCR amplification and is therefore highly quantitative. Unfortunately, this version of CAGE is different to that generated by the modENCODE project which used PCR amplification. This means that the CAGE method referenced in the introduction as performing best in a recent review, and the CAGE data used throughout the results section as a comparison are different protocols.

The authors should update the introduction and also make this distinction clear.

The authors should perform MAPCap to compare with (nAnTi)CAGE in a well characterized cell line where (nAnTi)CAGE data is available or can be generated.

*<https://www.ncbi.nlm.nih.gov/pubmed/24927836>

2) FigS1d shows CAGE to have much higher sensitivity and precision than MAPCap in detecting TSS. Is this due to the superiority of CAGE, or were the modENCODE CAGE data used in defining TSS in the ensembl annotation.

The authors should check this and if this is the case an annotation predating or not defined with the CAGE data should be used.

If DNA accessibility or active histone marks are available for these tissues they could be used as orthogonal evidence for transcription in comparing the methods.

FigS1d also shows MAPCap with similar sensitivity to RAMPAGE, but lower precision, does this suggest that MAPCap has a higher false positive rate? Can this be explained?

3) The correlation between RAMPAGE and RNAseq should be added, to contrast the CAGE and MAPCap with Fig1d,e to demonstrate if accuracy of gene expression estimates compared to RNAseq is an advantage for MAPCap.

4) The authors give handling low input samples, in combination with sample multiplexing as an advantage of MAPCap. What are the minimum or recommended RNA input requirements and how do they compare with other methods.

5) The introduction states that "the variability of promoter usage and expression of different transcripts (such as ncRNAs, eRNAs etc.) has not been investigated" in dosage compensation. However, ncRNA and eRNA were excluded from the DE analysis performed in the MLE mutant

investigation of dosage compensation by counting just reads overlapping 5'UTRs.
If DE analysis of ncRNA is possible with MAPCap / icetea this could be demonstrated.

6) In the abstract "high-resolution detection of transcription start-sites and differential expression analysis in a single setup, using a fast and simple protocol" is misleading considering the protocol requires multi-step, multi-day procedure. The authors should be more precise in the description of the method. And provide more detail including initial RNA amount, how is the quantity and quality of RNA change during the steps, as it includes multiple rounds of heating, sonication, and incubation periods. What is the mapping rate to different genomic regions (5', intergenic, intronic, etc) compare to other methods?

7) In page 6, the authors state that "the s-oligo incorporates the sequences of both standard sequencing adapters which omits the usage of an RT-primer and allows for a highly efficient intramolecular ligation." The authors should describe more in detail how the s-oligo is constructed and the nature of the oligo: RNA based or DNA based? If DNA based, please describe the ligation efficiency between s-oligo to RNA as this process is known to be highly inefficient. If RNA-based, is it complete or partial modification on the oligo? While the RT-primer is not used, reverse transcription is still taking place and it is surprising that G addition is not observed in the MAPCap protocol. Can authors explain how and why G addition is omitted as compared to other protocols?

8) The authors state in page 9 that "replicate-based analysis increases sensitivity and robustness of TSS detection." But isn't this to be expected when two libraries are combined instead of looking at one library at a time? Please explain in more detail the strength of this approach when considering the same read (depth) coverage with other protocols and provide statistics to state that the method yields more robust TSS detection.

9) The authors state in page 11 that "an in-vivo high resolution analysis comparison different promoters has not been done before" which is clearly not true and many in vivo based analysis using CAGE have been reported. This in particular shows allele-specific usage of TSS in zebrafish: <https://www.ncbi.nlm.nih.gov/pmc/articles/PMC4820030/>

10) In addition to Spike-in normalization, it would be relevant to compare up/down regulated genes after more conventional approaches to data normalization methods such as TMM, DESEQ, RC, etc.

Minor Points

Peak calling with paraclu is often performed on pooled reads from all samples, the authors should also add this comparison to Fig 2a.

The authors should specify the procedure used for 'internal normalization'.

Instead of a single gene in Fig1f a metagene plot or binned signal along transcripts comparing the three methods could better demonstrate 5' specificity. Similarly, how does the proportion of reads mapping to TSS differ between the methods.

Are replicates available for either the CAGE or RAMPAGE to compare with Fig S1c?

Accession numbers for external datasets used should be added (CAGE, RAMPAGE, RNAseq).

The label 'modENCODE', could be changed to CAGE in Fig1f, S1d for consistency with the other figures.

Greek characters delta and mu are not inserted leaving boxes.

Reviewer #3 (Remarks to the Author):

In the present study, Bhadwaj et al. describe a new quantitative method called 'MAPCap' for detecting transcription start sites at high spatial resolution as well as gene expression levels at the same time. MAPCap combines affinity purification of mRNAs containing a 5' Cap with the sequencing library preparation of the FLASH protocol, an approach that was previously developed by the same group (Aktas et al., *Nature*, 2017). The MAPCap sequencing library preparation relies on random barcodes which allow the identification and computationally removal of PCR duplicates, and also includes spike-in controls to quantify changes in gene expression. Additionally, the authors provide a new R/Bioconductor package called 'icetea' for the computational analysis of MAPCap data. The authors apply the new method to developing fly embryos and larvae and observe developmental stage- and sex-specific TSS activities. By using mutants of the maleless RNA helicase (MLE), which is important for balancing X chromosomal gene dosage between male and female flies, the authors provide evidence for a global 2-fold downregulation of TSS activity on the X-chromosome.

Although, a set of protocols are available for determining the exact genomic locations of TSSs these methods usually do not allow the accurate quantification of gene expression levels. MAPCap overcomes this limitation by combining the purification of capped mRNAs with a state-of-the-art sequencing library preparation method that also includes spike-in RNA controls. I have no doubts that this protocol will be broadly applicable because it allows TSS and differential gene expression to be measured at the same time and due to the computational analysis package that is provided for MAPCap data. The new method and the major findings of this study will be of broad interest for researchers in the transcription field and for colleagues in other research areas with a general interest in gene regulation.

Here are some comments that need to be addressed prior to publication.

Major comments:

(1) Several protocols are available to determine TSS positions at high resolution such as CAGE, RAMPAGE, GRO-cap and others. The authors now present a new approach for profiling TSSs at high resolution. A question that immediately arises in the first section of the Results section is how reproducibly TSS locations are detected by MAPCap between biological replicates? The authors should provide this information early on, for instance at the beginning of the second paragraph (page 6).

(2) Along these lines, the authors also compare MAPCap data with data obtained from other TSS profiling methods such as CAGE. On page 7 (second paragraph) the authors claim that 'MAPCap signal shows good correlation with other protocols'. The authors should mention the correlation coefficients between the different methods here. New figures that result from this analysis can be added to the supplement.

(3) The authors claim that MAPCap is 'fast and easy to perform'. This statement is vague. RAMPAGE takes two days. How long does MAPCap take? What does 'easy' exactly mean here? Less experimental steps? A detailed comparison with the experimental steps of CAGE, RAMPAGE and GRO-cap would help. Along these lines, the authors should highlight in Figure 1a the new steps of the MAPCap approach and should clearly label the steps that were adapted from pre-existing methods such as from the FLASH protocol (mainly the sequencing library prep).

(4) MAPCap as compared to CAGE and RAMPAGE shows a relatively low precision (~ 0.5) using paraclu (Figure S1d). This means that 1 out of 2 identified TSSs is a false positive. This needs to be discussed in the manuscript. How is the sensitivity and precision improving when the 'local enrichment' algorithm that is implemented in 'icetea' is applied?

Minor comments:

(1) Regarding the description of the MAPCap library preparation in the methods section: please add the unit number for each enzyme that is used, including for RNase H, Terminator exonuclease etc., and also mention how much s-oligo was used for ligation. This information is essential for potential future users of this approach.

(2) The authors provide evidence that a removal of PCR duplicates in the absence of random barcodes lead to near-complete loss of signal in case of the CAGE data, as expected (Fig. 1f). This is shown for only one (Chc) gene. The authors should provide evidence, if this is also true on a global scale.

(3) On page 3 second paragraph and on page 7 third paragraph the authors mention that RAMPAGE relies on 'pseudo-random barcodes'. What is the difference to 'real' random barcodes? Please clarify in the main text.

(4) Page 9, second paragraph: what do the authors mean by 'long TSSs'? Extended genomic region with multiple TSSs? This needs to be clarified in the main text.

(5) The font sizes need to be increased in almost all figures, including the supplementary figures. Labels and legends are sometimes missing (e.g. Figure 3d, h) and genome tracks are lacking gene labels (e.g. Figure 2d, e and Figure 3e, g). This needs to be fixed.
Figure 1b: the color between CAGE (dark blue) and RAMPAGE (light blue) can hardly be distinguished. Please change.

(6) The manuscript contains several typos that need to be fixed such as on:

(1) page 3 end of second paragraph: 'could be further improved by the use biological...' should be replaced by 'could be further improved by the use of biological...'

(2) page 6 end of first paragraph: 'we an easily control' by 'we can easily control'

(3) page 10 first paragraph: '1574 enhancers RNAs' by '1574 enhancer RNAs'

NCOMMS-19-04815-T

Bhardwaj et al.

Point by point response to reviewers

Summary:

- We provide a detailed, step-by-step MAPCap protocol in the Nature Protocols format (as reviewer's file, also uploaded to protocol exchange) that would simplify its widespread adoption.
- We have expanded our comparison of MAPCap with external protocols by adding additional statistics and evidence for the TSS, such as ChIP-seq datasets of active marks and DNaseq-seq data in embryos.
- We added a new MAPCap experiment in mouse ESCs, comparing it with nAnTiCAGE as well as demonstrating the allele-specific detection of TSS.
- We added a new MAPCap experiment in S2 cells to show the TSS enrichment at various concentrations of starting RNA (up to 100ng).
- We show that icetea provides superior results using replicates even on external datasets, using CAGE and DNaseq-seq analysis in S2 cells.
- We expanded our analysis of stage and sex-specific eRNAs by adding online CAGE data in adult male and female heads and comparing to our embryo and larvae MAPCap.

Finally, we added various clarifications and comparison in the text, expanded the figure sizes and answered all other individual issues raised by the reviewers. All these changes are highlighted in yellow.

Reviewer #1

Overall, Akhtar and colleagues have combined existing methodologies to develop a novel MAPcap method which is improved because it allows quantitative comparisons not possible with prior approaches for quantifying TSS. They have used early multiplexing of samples, removed PCR duplicates by random barcodes and used external spike in controls to allow accurate quantification of TSS expression. They quantified X-chromosome dosage compensation and discovered that MLE has an interesting sex-specific

role in the brain. The authors also confirmed prior observations that different roX promoters are developmentally regulated.

We thank the reviewer for the supportive comments and finding our method novel.

There are several important concerns to address:

1) The authors measure dosage compensation without looking at elongation which is a caveat that should be mentioned in the text.

Indeed, MAPCAP is specifically designed to assess the transcription starts by mapping transcription start sites. We have now elaborated on this point and indicated which methods will be suitable to calculate initiation and elongation rates (**page 19**).

2) All figures are very small and many are missing axis labels. Please recheck all labels and increase the size of figures.

Apologies for the inconvenience. We have expanded the font sizes in the figures and placed labels wherever required.

3) I would like to see more details about their computational icetea method. I read their tutorial on Bioconductor but I do not think they provided enough detail in this paper.

We have now added details of the methods implemented in the icetea in a new section “TSS analysis methods implemented in the icetea package” under methods (**page 32-33**).

4) The authors have shown enhancer RNAs that have stage specific expression but they should have determined whether eRNAs also showed sex-specific expression. Were there no sex-specific eRNAs (Fig. 2g)? This should be mentioned and commented on.

Indeed, we analyzed the difference in eRNAs between sexes and detected sex-specific eRNAs (**Fig. 2h, Supplementary Fig. 4e**). We note one caveat, that these eRNAs show high variability in expression between biological replicates and therefore it's difficult to conclude whether a majority of them are truly sex-specific (**Supplementary Fig. 5d**). Therefore, we focussed more on the stage-specific eRNAs in the manuscript.

5) Embryo and sexed larval brains is not the best comparison because embryo is both male and female and brain is a specific tissue while embryo represents a whole organism. It would be much better if larval and adult brain could be compared in males and females separately.

We thank the reviewer for the suggestion. Upon revision, we have now expanded the analysis (**Fig. 2g-h, Supplementary Fig. 4b-d**) to include CAGE data from male and female adult heads. Most eRNAs are stage-specific rather than sex-specific. However, we agree that it would be interesting to investigate the tissue specificity of eRNA expression using MAPCap by doing a larger tissue-by-tissue comparison in a future study.

6) Does MLE have any role in females or on autosomes? This should be addressed.

There might be some misunderstanding by the reviewer. Indeed, we did investigate the role of MLE in females and on autosomes. We find that genes with sexually dimorphic expression are regulated by MLE in both, males and females, and many of them are also on autosomes. This points to a novel role of MLE in regulating an X-independent process in both sexes (**Page 13-14, Fig. 3f, Supplementary Fig. 4e-h**)

Specific comments:

Their materials and methods missed some detailed information such as the stage and number of embryos used.

Thanks for pointing this out. We have added these details in the revised version (**page 22**).

Fig 1c-e, What normalization of read counts did they use when calculating the correlation between CAGE vs. MAPCap, and compare them to RNA-seq. They didn't show units on X- Y- axes.

We used log(Counts per million) of reads for all these plots (with a pseudocount of 1 to avoid the log of zeros). We updated **Fig. 1c-d, Supplementary Fig. 1** and the corresponding legend to mention this.

Page 9: why brain tissue, why not study in sexed embryos? Why this stage? is this the stage with brain developed?

Sexual dimorphism in the brain leads to sex specific behaviour in flies, such as male courtship behaviour. Although structural and transcriptomic studies have been performed on the brain, an analysis of promoters usage has been lacking. Therefore, we thought it serves as a nice system to perform MAPCap, as our data shall also serve as a useful resource for future studies.

Although cells that contribute to brain start differentiating in early embryos, a functional and dissectable brain can only be obtained in larval stage. We specifically chose L3 larvae as this is the stage where defects in dosage compensation lead to male lethality. This also gives us an opportunity to understand the role of MLE in sexual dimorphism, as we discussed in the manuscript (**Page 13-14**).

We have clarified these rationale in the revised version (**page 10**).

Page 12 top paragraph: how similar the DEGs are between RNA-seq and MAPCap? how much in percentage? Any Venn diagram? Did they only used DEGs that presented in both method? Any GO annotation/pathway analysis for the DEGs?

Along with the concordance of differential expression estimated provided before (**Fig. 3d, supplementary Fig. 5a**), we have now added the GO term analysis and venn diagrams in **Supplementary Fig. 5b-c**. We used DEGs from both methods for the comparison of wild-type males and females, while we used DEGs from MAPCap for Wild-type: mutant comparison.

Page 13 top paragraph: MLE mutants caused both autosomal gene and X-linked gene to change (Fig 3f), so the calculation of the X:A ratio in mutant males and females may not be appropriate.

We only calculated X:A ratio in mutants as we were curious to compare it with the X:A dosage compensation estimates presented in previous studies (which used other orthogonal approaches). For all other analysis, we have only used X:X ratios between sexes.

It would be better to plot a histogram of the changes on the X and the changes and autosomes and overlay them.

We have added this in the revised version (**Fig. 4a**).

Page 13 bottom paragraph: How do the MSL sensitive vs. insensitive genes that they identified compare to other studies such as Hamada et al. 2004, Lott and Eisen, or Brian Oliver's studies in adult heads?

We have added the comparison with Lott and Eisen (2011) study that studies the difference in gene expression between sexes during the onset of dosage compensation in early embryos (**Supplementary Fig. 6d, page 15**). We find that MLE-sensitive genes identified in our study seem to be tightly dosage compensated while the MLE-insensitive genes have a compensation score (male:female slope ratio) similar to autosomes. We couldn't download the gene list from Hamada et. al. (dead links). The study from Chen and Oliver (2015) investigates whether X and autosomal genes respond similarly to gene dosage on selected regions with induced hemizyosity (microdeletions, avoiding the MSL complex machinery) compared to wild-type. Therefore this data does not indicate MSL sensitivity of genes. However the sex-bias genes identified in their study do show similar sex bias in our study (figure attached below).

Fig 4a. is t-test a correct test to compare the expression distribution? A K-S test is likely more appropriate to define the difference between two distributions.

We had performed t-test assuming a normal/log-normal distribution of gene expression. During the revision, we checked for normality of both expression and distance to nearby features using

shapiro-wilk test. Results suggest that the assumption is correct ($p < 2.2e-16$). A KS test however, also indicates a significant difference ($P \sim 0$).

Page 17: Can the effects of genomic imprinting be studied using MAPCap? Does it have enough coverage to call the allelic-specific expression at TSS?

While addressing reviewer #2 comment #1, we performed a proof-of-principle MAPCap experiment in mouse ESCs with dual-hybrid background and checked how much signal remains after allele-specific sorting of MAPCap data. We find that 16% of MAPCap reads had unique allele-specific signal. While it is smaller than what we obtain with RNA-seq (around 25%), MAPCap libraries sequenced with appropriate depth (>5Mil) can easily be used for allele-specific TSS detection (**Supplementary Fig 2d; page 7, top paragraph**). Therefore, we believe that MAPCap would be indeed appropriate to study imprinting and allele-specific expression at TSS. We are interested in further optimizing and applying MAPCap to study these processes in mammals in the future.

Reviewer #2

The authors develop MAPCap, a novel method for sequencing capped transcript 5' ends incorporating s-oligo with unique molecular identifiers (UMI) and sample multiplexing. MAPCap is then performed in Drosophila embryos and compared to the CAGE and RAMPAGE protocols. The authors also implement a clustering method to identify TSS which is compared to the previous 'paraclu' method, with advantages when sample replicates are available. Sex specific TSS, and transcribed enhancer loci during Drosophila development are identified. MAPCap is also performed in MLE mutant Drosophila to examine gene dosage. The authors provide their bioinformatic analysis workflow, and an R package (icetea) for the analysis of MAPCap data.

We thank the reviewer for acknowledging the novelty of our method.

Main Points

1) The introduction states "Quantification of transcript expression from CAGE is hindered by the PCR amplification bias". This is true for some CAGE methods including nanoCAGE and C1-CAGE, however cap-trapping (nAnTi)CAGE which performed best in the recent comparative review of 5p profiling*

methods referenced in the introduction uses no PCR amplification and is therefore highly quantitative. Unfortunately, this version of CAGE is different to that generated by the modENCODE project which used PCR amplification. This means that the CAGE method referenced in the introduction as performing best in a recent review, and the CAGE data used throughout the results section as a comparison are different protocols.

The authors should update the introduction and also make this distinction clear.

Thanks for pointing out the difference. We have clarified this distinction in the introduction (**page 2**).

The authors should perform MAPCap to compare with (nAnTi)CAGE in a well characterized cell line where (nAnTi)CAGE data is available or can be generated.

[*https://www.ncbi.nlm.nih.gov/pubmed/24927836](https://www.ncbi.nlm.nih.gov/pubmed/24927836)

Upon reviewer's request, we have now performed an initial MAPCap in mouse ESCs, where nAnTiCAGE data is available from a recent study (Cvetesic et. al. 2018), and compared the two datasets (**Fig. 1f, Supplementary Fig. 2b-c; page 7, top paragraph**). The correlation is lower than our initial expectation. However, this is very likely be due to 1) a technical issue: low material left after cap enrichment, leading to higher PCR cycles (18 cycles) 2) the two cell lines are not exactly of the same background. We strongly believe that the correlation can be further improved if both the protocols are systematically performed in parallel on the same cell line. This, however, deserves an independent benchmarking study which is beyond the scope of the current manuscript. Nonetheless, we keep this new analysis in the manuscript, as it provides a "proof of principle" experiment to show the versatility of MAPCap use in mammalian cells.

2) FigS1d shows CAGE to have much higher sensitivity and precision than MAPCap in detecting TSS. Is this due to the superiority of CAGE, or were the modENCODE CAGE data used in defining TSS in the ensembl annotation.

The authors should check this and if this is the case an annotation predating or not defined with the CAGE data should be used.

We agree with the reviewer and this is indeed the case. We used the ENSEMBL (release 79) annotation of the TSS for this analysis. For *Drosophila*, both ensembl and UCSC annotations are in fact, obtained from Flybase :

- http://mar2015.archive.ensembl.org/Drosophila_melanogaster/Info/Annotation
- <http://hgdownload.cse.ucsc.edu/goldenPath/dm6/database/>

Since 2010, the Flybase makes use of the TSS mapping data from modENCODE and other TSS mapping projects (including RAMPAGE) to update its annotation, and therefore using any (UCSC/Ensembl/Flybase) annotation of the dm6 assembly would produce results in favor of the CAGE and RAMPAGE protocols:

- https://wiki.flybase.org/wiki/FlyBase:Gene_Model_Annotation_Guidelines.
- <http://www.g3journal.org/content/5/8/1721.long>

We therefore do not expect to perform better in these metrics when using any dm6 annotation as a gold standard but rather aim to achieve comparable results.

If DNA accessibility or active histone marks are available for these tissues they could be used as orthogonal evidence for transcription in comparing the methods.

We thank the reviewer for the suggestion. Upon revision, we have now used the modENCODE ChIP-seq data for active histone marks (H3K4me1, H3K4me3 and H3K27ac) and DNase-seq data from a comparable stage as additional evidence for an active TSS (**Supplementary Fig. 3b**). We find that about 1/5th of the TSSs classified as “false positives” have additional evidence of active transcription.

FigS1d also shows MAPCap with similar sensitivity to RAMPAGE, but lower precision, does this suggest that MAPCap has a higher false positive rate? Can this be explained?

In the previous analysis we plotted the metrics (precision/sensitivity and F1-score) at the minimal peak detection threshold from paraclu; signal enrichment (density rise) ≥ 1 , number of reads in peaks ≥ 1 , which lead to a high number of false positives in the MAPCap data. In the new comparison, we optimized these parameters to achieve the maximum F1-score for each protocol independently (similar to Adiconis et al., 2018). We also utilized the DHS peaks to

detect false negatives. This improves the precision-sensitivity metric for MAPCap (revised **Supplementary Fig. 3a**). Further, adding additional evidence as suggested by the reviewer, we find that 1/5th of “false positive” peaks in MAPCap have additional supportive evidence (**Supplementary Fig. 3b**). We would therefore suggest that MAPCap performs comparable to RAMPAGE and CAGE in TSS detection.

3) The correlation between RAMPAGE and RNAseq should be added, to contrast the CAGE and MAPCap with Fig1d,e to demonstrate if accuracy of gene expression estimates compared to RNAseq is an advantage for MAPCap.

We now added this in **Supplementary Fig. 1h**. We have further added the PCA plot in **Supplementary Fig. 1f** to show the comparison of all 3 protocols in the same plot.

4) The authors give handling low input samples, in combination with sample multiplexing as an advantage of MAPCap. What are the minimum or recommended RNA input requirements and how do they compare with other methods.

In order to show the effect of low input samples on peak calling and signal enrichment, we added data from a MAPCap experiment on wild-type S2 cells with a range of starting material. MAPCap shows good enrichment upto 100ng total RNA as starting material, but in absence of replicates, we recommend starting with about 500ng to avoid false positive peaks (**Fig. 1g**, **Supplementary Fig. 2 e**). We have discussed these results on **page 7** and the comparison with other protocols under the discussion section (**page 16-17**).

5) The introduction states that “the variability of promoter usage and expression of different transcripts (such as ncRNAs, eRNAs etc.) has not been investigated” in dosage compensation. However, ncRNA and eRNA were excluded from the DE analysis performed in the MLE mutant investigation of dosage compensation by counting just reads overlapping 5'UTRs.

If DE analysis of ncRNA is possible with MAPCap / icetea this could be demonstrated.

Due to counting reads on 5'UTRs, non-coding RNAs were excluded for comparison of RNA-seq and MAPCap in wild-type Males vs Females. However, the analysis of MLE mutants and

dosage compensation was performed on all TSSs, including ncRNAs. Upon revision, we have additionally discussed how lncRNAs and eRNAs on X chromosome respond in MLE mutants to address the reviewer's point (**page 14, Fig. 4b**).

6) In the abstract "high-resolution detection of transcription start-sites and differential expression analysis in a single setup, using a fast and simple protocol" is misleading considering the protocol requires multi-step, multi-day procedure. The authors should be more precise in the description of the method. And provide more detail including initial RNA amount, how is the quantity and quality of RNA change during the steps, as it includes multiple rounds of heating, sonication, and incubation periods. What is the mapping rate to different genomic regions (5', intergenic, intronic, etc) compare to other methods?

We agree that a more detailed description of MAPCap protocol would be necessary for its broad adoption. We therefore provide a detailed, step-by-step MAPCap protocol in this revision (**reviewer's file**, which we have also uploaded on the Nature Protocols exchange platform), mentioning the time required for each step and RNA quality plots. Mapping rates are shown in **Supplementary Fig. 2b**.

7) In page 6, the authors state that "the s-oligo incorporates the sequences of both standard sequencing adapters which omits the usage of an RT-primer and allows for a highly efficient intramolecular ligation." The authors should describe more in detail how the s-oligo is constructed and the nature of the oligo: RNA based or DNA based? If DNA based, please describe the ligation efficiency between s-oligo to RNA as this process is known to be highly inefficient. If RNA-based, is it complete or partial modification on the oligo? While the RT-primer is not used, reverse transcription is still taking place and it is surprising that G addition is not observed in the MAPCap protocol. Can authors explain how and why G addition is omitted as compared to other protocols?

We have added more details of the s-oligo and G-addition (**page 5-6**) under the "Results" section. The s-oligo is an RNA-DNA hybrid where the RNA-part is used for ligation and the DNA part serves as the pre-designed template for reverse transcription. Further details on the design of s-oligo would also appear in our upcoming manuscript (under review) where we studied several human RNA-binding proteins using FLASH. We therefore describe the design here in brief in order to avoid redundancy.

The SuperScript-III reverse transcriptase (used in MAPCap) has intrinsically low terminal nucleotide transferase (TdT) activity, which, in fact, depends on the concentration of enzyme, time of incubation, temperature and buffer condition ^{1,2}. Cap-trapping methods enhance this by changing the buffer concentration of Mn⁺ and Mg⁺ ions ³, while template-switching protocols further maximize this by additionally providing the non-template oligos ⁴. In MAPCap, we simply skip these optimization steps as we do not rely on them for selection of cDNA Cap, therefore reducing the 'G' addition bias.

8) The authors state in page 9 that “replicate-based analysis increases sensitivity and robustness of TSS detection.” But isn't this to be expected when two libraries are combined instead of looking at one library at a time? Please explain in more detail the strength of this approach when considering the same read (depth) coverage with other protocols and provide statistics to state that the method yields more robust TSS detection.

Combination of replicates can be performed by pooling (merging of reads) or intersection (taking common TSS) of replicates. In the revised version, we have further added the comparison of our algorithm with these two alternative approaches, and used AUPRC (area under precision-recall curve) as the performance metric. Our method provides better results than these alternatives (Revised **Fig. 2a-b**).

Our algorithm is protocol-agnostic and can also be applied to other protocols with replicates to improve accuracy of TSS detection. In the revised manuscript, we show the results with 1 Million subsampled reads for both MAPCap and CAGE data. In case of CAGE data, we obtain many more true positive peaks while keeping the same optimal F1-score as with paraclu (**Fig. 2b and Supplementary figures 3d-f, page 7-8**).

9) The authors state in page 11 that “an in-vivo high resolution analysis comparison different promoters has not been done before” which is clearly not true and many in vivo based analysis using CAGE have been reported. This in particular shows allele-specific usage of TSS in zebrafish: <https://www.ncbi.nlm.nih.gov/pmc/articles/PMC4820030/>

We referred to this in the context of dosage compensation in flies. We have now clarified this in the text (**page 13**).

10) In addition to Spike-in normalization, it would be relevant to compare up/down regulated genes after more conventional approaches to data normalization methods such as TMM, DESEQ, RC, etc.

We have added a comparison of our results with those obtained from TMM and DESeq normalization in terms of overall numbers, direction and fraction of affected TSS on X-chromosome. These alternative normalization methods produced a balanced set of Up and Down-regulated genes and masked a global expression shift on the X-chromosome. We added the comparison in **Supplementary Fig. 5i** and expanded the text (**page 13**).

Minor Points

Peak calling with paraclu is often performed on pooled reads from all samples, the authors should also add this comparison to Fig 2a.

We have now expanded **Fig. 2a** to compare our results with paraclu on “pooled” samples, as well as on individual samples followed by taking an “intersection” of results. Indeed, the icetea method provides better accuracy in TSS detection than either of these approaches.

The authors should specify the procedure used for ‘internal normalization’.

In the original submission, we used DESeq2 method as “internal normalization” to compare with our spike-in results, which we have now expanded to include multiple methods (**page 14**). We provide multiple normalization methods such as TMM, DESeq2 etc. in the icetea package, which we have now mentioned in the extended description of the icetea package under “methods” (**page 32-33**).

Instead of a single gene in Fig1f a metagene plot or binned signal along transcripts comparing the three methods could better demonstrate 5' specificity. Similarly, how does the proportion of reads mapping to TSS differ between the methods.

We have now provided a metagene profile of the different protocols in **Supplementary Fig. 2a**, which shows 5'-specificity as well as the preservation of signal after removal of PCR duplicates

in the MAPCap protocol. We typically find that >60% of duplicate-free reads in our MAPCap experiments are within TSS (**Supplementary Fig. 4f**). This is close to CAGE/RAMPAGE (70-80%) and better than low input protocols such as slicCAGE/nanoCAGE. However, this can further be increased if the TSS detection using icetea is optimized.

Are replicates available for either the CAGE or RAMPAGE to compare with Fig S1c?

Replicates are available for the CAGE data in S2 cells from modENCODE, we have processed them using the same parameters as MAPCap and added them for comparison (both, before and after PCR de-duplication) in **Supplementary Fig. 1e-f**.

Accession numbers for external datasets used should be added (CAGE, RAMPAGE, RNAseq).

We added accession numbers for all external datasets as a Table on **page 25**.

The label 'modENCODE', could be changed to CAGE in Fig1f, S1d for consistency with the other figures.

Greek characters delta and mu are not inserted leaving boxes.

We replaced the figure labels and fixed the issue with greek letters.

Reviewer #3

In the present study, Bhadwaj et al. describe a new quantitative method called 'MAPCap' for detecting transcription start sites at high spatial resolution as well as gene expression levels at the same time. MAPCap combines affinity purification of mRNAs containing a 5' Cap with the sequencing library preparation of the FLASH protocol, an approach that was previously developed by the same group (Aktas et al., Nature, 2017). The MAPCap sequencing library preparation relies on random barcodes which allow the identification and computationally removal of PCR duplicates, and also includes spike-in controls to quantify changes in gene expression. Additionally, the authors provide a new R/Bioconductor package called 'icetea' for the computational analysis of MAPCap data. The authors apply the new method to developing fly embryos and larvae and observe developmental stage- and sex-specific TSS activities. By

using mutants of the maleless RNA helicase (MLE), which is important for balancing X chromosomal gene dosage between male and female flies, the authors provide evidence for a global 2-fold downregulation of TSS activity on the X-chromosome.

Although, a set of protocols are available for determining the exact genomic locations of TSSs these methods usually do not allow the accurate quantification of gene expression levels. MAPCap overcomes this limitation by combining the purification of capped mRNAs with a state-of-the-art sequencing library preparation method that also includes spike-in RNA controls. I have no doubts that this protocol will be broadly applicable because it allows TSS and differential gene expression to be measured at the same time and due to the computational analysis package that is provided for MAPCap data. The new method and the major findings of this study will be of broad interest for researchers in the transcription field and for colleagues in other research areas with a general interest in gene regulation.

We thank the reviewer for appreciating the usefulness of our study and for the encouraging remarks.

Here are some comments that need to be addressed prior to publication.

Major comments:

(1) Several protocols are available to determine TSS positions at high resolution such as CAGE, RAMPAGE, GRO-cap and others. The authors now present a new approach for profiling TSSs at high resolution. A question that immediately arises in the first section of the Results section is how reproducibly TSS locations are detected by MAPCap between biological replicates? The authors should provide this information early on, for instance at the beginning of the second paragraph (page 6).

We thank the reviewer for this comment. Upon revision, apart from the correlation on known 5'-UTRs, we now also provide correlation between replicates on the detected TSS. For comparison, we also calculated similar correlation on modENCODE CAGE data for S2 cells with replicates, using identical approach, with and without PCR duplicates (see methods; page). MAPCap shows a consistent 90-94% correlation between replicates on previously known 5'UTRs as well as on detected TSSs. CAGE also performs at the similar range, although the duplicate-free correlation is relatively lower. We added these results in **Supplementary Fig. 1d-f** and in the second paragraph of our results section (**page 5**), as suggested.

(2) Along these lines, the authors also compare MAPCap data with data obtained from other TSS profiling methods such as CAGE. On page 7 (second paragraph) the authors claim that 'MAPCap signal shows good correlation with other protocols'. The authors should mention the correlation coefficients between the different methods here. New figures that result from this analysis can be added to the supplement.

We have added the correlation coefficients in the text (**page 6**), and have further expanded the **Supplementary Fig. 1** to add scatter plots and also a PCA plot to show relationship between the methods.

(3) The authors claim that MAPCap is 'fast and easy to perform'. This statement is vague. RAMPAGE takes two days. How long does MAPCap take? What does 'easy' exactly mean here? Less experimental steps? A detailed comparison with the experimental steps of CAGE, RAMPAGE and GRO-cap would help. Along these lines, the authors should highlight in Figure 1a the new steps of the MAPCap approach and should clearly label the steps that were adapted from pre-existing methods such as from the FLASH protocol (mainly the sequencing library prep).

Thanks for pointing this out. We agree that a more detailed description of MAPCap protocol would be necessary for its broad adoption. In this revision, we provide the full MAPCap protocol with details of each step, the time required, critical steps and QC plots (**Reviewer's file**, which we have also uploaded on the Nature Protocols exchange platform). While the ease of use can only be judged by independent users, MAPCap minimizes the number of critical or difficult steps. MAPCap is the fastest amongst the TSS-profiling protocols, requiring 16 hours in total for 12 or more samples. We have updated the introduction (**page 3**) and discussion (**page 16**) to mention this. The MAPCap method is performed entirely different from CAGE/RAMPAGE and GRO-Cap, which can be seen by comparing our detailed submitted protocol with other published protocols. Furthermore, we have modified **Fig. 1a** to highlight the steps which are common between the MAPCap and FLASH protocol.

(4) MAPCap as compared to CAGE and RAMPAGE shows a relatively low precision (~ 0.5) using paraclu (Figure S1d). This means that 1 out of 2 identified TSSs is a false positive. This needs to be discussed in the manuscript. How is the sensitivity and precision improving when the 'local enrichment' algorithm that is implemented in 'icetea' is applied?

Upon revision, we further investigated into the “false positives” from the MAPCap protocol. After taking the maximal F1-score for each protocol independently, the gap in sensitivity between MAPCap and other protocols reduces (please also see reply to comment #2 from Reviewer #2). We then looked for further supportive evidence using DNase-seq and ChIP-seq data on these false positive TSSs and find that 1/5th of the False positive TSS have orthogonal supporting evidence. We have added these results in the manuscript (**page 8-9; Supplementary Fig. 3a-b**). We have also plotted the precision-recall curve and added the AUPRC values (**Fig 2b**) to show how the sensitivity and precision improve when icetea is applied to the data.

Minor comments:

(1) Regarding the description of the MAPCap library preparation in the methods section: please add the unit number for each enzyme that is used, including for RNase H, Terminator exonuclease etc., and also mention how much s-oligo was used for ligation. This information is essential for potential future users of this approach.

We agree with the reviewer that these details are important. Our step-by-step protocol (**reviewer’s file**) includes the concentration/units of all required reagents.

(2) The authors provide evidence that a removal of PCR duplicates in the absence of random barcodes lead to near-complete loss of signal in case of the CAGE data, as expected (Fig. 1f). This is shown for only one (Chc) gene. The authors should provide evidence, if this is also true on a global scale.

We have added the metagene profiles showing TSS signal between protocols on all genes after removing PCR duplicates (**Supplementary Fig. 2a**).

(3) On page 3 second paragraph and on page 7 third paragraph the authors mention that RAMPAGE relies on ‘pseudo-random barcodes’. What is the difference to ‘real’ random barcodes? Please clarify in the main text.

Unlike MAPCap, the RAMPAGE protocol does not use pre-designed random barcodes. Instead, the protocol relies on the alignment mismatch between the PCR primer and the genome to perform de-duplication. We have mentioned this now in the discussion of PCR bias removal under the results section (**page 6**).

(4) Page 9, second paragraph: what do the authors mean by 'long TSSs'? Extended genomic region with multiple TSSs? This needs to be clarified in the main text.

We actually used “long TSSs” exchangeably with “broad TSSs” in that paragraph. We apologise for the confusion and have replaced “long TSSs” with “broad TSSs” in the revised version.

(5) The font sizes need to be increased in almost all figures, including the supplementary figures. Labels and legends are sometimes missing (e.g. Figure 3d, h) and genome tracks are lacking gene labels (e.g. Figure 2d, e and Figure 3e, g). This needs to be fixed.

Figure 1b: the color between CAGE (dark blue) and RAMPAGE (light blue) can hardly be distinguished. Please change.

We have increased the font size for all figures, placed axis labels and gene names, and also updated **Fig. 1b** to better distinguish the colors.

(6) The manuscript contains several typos that need to be fixed such as on:

(1) page 3 end of second paragraph: 'could be further improved by the use biological...' should be replaced by 'could be further improved by the use of biological...'

(2) page 6 end of first paragraph: 'we an easily control' by 'we can easily control'

(3) page 10 first paragraph: '1574 enhancers RNAs' by '1574 enhancer RNAs'

We have fixed the above along with other typos that we could identify in the manuscript.

References

1. Sterling, C. H., Veksler-Lublinsky, I. & Ambros, V. An efficient and sensitive method for preparing cDNA libraries from scarce biological samples. *Nucleic Acids Res.* **43**, e1 (2015).
2. Chen, D. & Patton, J. T. Reverse transcriptase adds nontemplated nucleotides to cDNAs during 5'-RACE and primer extension. *Biotechniques* **30**, 574–80, 582 (2001).
3. Schmidt, W. M. & Mueller, M. W. CapSelect: a highly sensitive method for 5' CAP-dependent enrichment of full-length cDNA in PCR-mediated analysis of mRNAs. *Nucleic Acids Res.* **27**, e31 (1999).
4. Pinto, F. L. & Lindblad, P. A guide for in-house design of template-switch-based 5' rapid amplification of cDNA ends systems. *Anal. Biochem.* **397**, 227–232 (2010).

REVIEWERS' COMMENTS:

Reviewer #1 (Remarks to the Author):

The authors have done a very impressive job revising the manuscript and adding much further detail and many new experiments across species.

All of my comments have been fully addressed.

We look forward to trying the MAPcap approach.

Reviewer #2 (Remarks to the Author):

The authors addressed all of the questions and I do not have additional points.

Reviewer #3 (Remarks to the Author):

The authors have addressed all of my comments. Most notably, the authors now (1) provide a step-by-step protocol for MAPCap, (2) provide new evidence for a high reproducibility between replicate samples (Fig. S1 d-f), (3) performed correlation analyses between MAPCap data and data obtained by other TSS profiling methods (Fig. S1 g-j), and (4) also provide new analyses on the precision and sensitivity of the MAPCap approach. Furthermore, Bhardwaj et al. now also show that MAPCap works robustly when different amounts of RNA are used as input for the MAPCap library preparation (Fig. 1g).

The additional experiments as well as the clarifications added to the main text strongly improve the manuscript.

REVIEWERS' COMMENTS

Reviewer #1

The authors have done a very impressive job revising the manuscript and adding much further detail and many new experiments across species.

All of my comments have been fully addressed.

We look forward to trying the MAPcap approach.

Reviewer #2

The authors addressed all of the questions and I do not have additional points.

Reviewer #3

The authors have addressed all of my comments. Most notably, the authors now (1) provide a step-by-step protocol for MAPCap, (2) provide new evidence for a high reproducibility between replicate samples (Fig. S1 d-f), (3) performed correlation analyses between MAPCap data and data obtained by other TSS profiling methods (Fig. S1 g-j), and (4) also provide new analyses on the precision and sensitivity of the MAPCap approach. Furthermore, Bhardwaj et al. now also show that MAPCap works robustly when different amounts of RNA are used as input for the MAPCap library preparation (Fig. 1g).

The additional experiments as well as the clarifications added to the main text strongly improve the manuscript.

We would like to thank all three reviewers for providing constructive and encouraging reviews that helped improve our manuscript.